# Port-Hamiltonian Architectural Bias for Long-Range Propagation in Deep Graph Networks

**Simon Heilig**[1*], **Alessio Gravina**[2*], **Alessandro Trenta**[2], **Claudio Gallicchio**[2], **Davide Bacciu**[2]

[1]Ruhr University Bochum, Department of Computer Science, Germany
[2]University of Pisa, Department of Computer Science, Italy

## Abstract

The dynamics of information diffusion within graphs is a critical open issue that heavily influences graph representation learning, especially when considering long-range propagation. This calls for principled approaches that control and regulate the degree of propagation and dissipation of information throughout the neural flow. Motivated by this, we introduce *port-Hamiltonian Deep Graph Networks*, a novel framework that models neural information flow in graphs by building on the laws of conservation of Hamiltonian dynamical systems. We reconcile under a single theoretical and practical framework both non-dissipative long-range propagation and non-conservative behaviors, introducing tools from mechanical systems to gauge the equilibrium between the two components. Our approach can be applied to general message-passing architectures, and it provides theoretical guarantees on information conservation in time. Empirical results prove the effectiveness of our port-Hamiltonian scheme in pushing simple graph convolutional architectures to state-of-the-art performance in long-range benchmarks.

## 1 Introduction

The conjoining of dynamical systems and deep learning has become a topic of great interest in recent years. In particular, neural differential equations (neural DEs) demonstrate that neural networks and differential equations are two sides of the same coin (Haber & Ruthotto, 2017; Chen et al., 2018; Chang et al., 2019). This connection has been pushed to the domain of graph learning (Bacciu et al., 2020; Wu et al., 2020), forging the field of differential-equations inspired Deep Graph Networks (DE-DGNs) (Poli et al., 2019; Chamberlain et al., 2021a; Gravina et al., 2023; Han et al., 2024; Gravina et al., 2025a).

In this paper, we are interested in designing the information flow within a graph as a solution of a port-Hamiltonian system (Van der Schaft, 2017), which is a general formalism for physical systems that allows for both conservative and non-conservative dynamics, with the aim of allowing flexible long-range propagation in DGNs. Indeed, long-range propagation is an ongoing challenge that limits the power of the Message-Passing Neural Network (MPNN) family (Gilmer et al., 2017), as their capacity to transmit information between nodes exponentially decreases as the distance increases (Alon & Yahav, 2021; Di Giovanni et al., 2023). This prevents DGNs from effectively solving real-world tasks, e.g., predicting anti-bacterial properties of peptide molecules (Dwivedi et al., 2022). While recent literature proposes various approaches to mitigate this issue, such as graph rewiring (Gasteiger et al., 2019; Topping et al., 2022; Gutteridge et al., 2023) and graph transformers (Shi et al., 2021; Dwivedi & Bresson, 2021; Wu et al., 2023), here we aim to address this problem providing a theoretically grounded framework through the prism of port-Hamiltonian-inspired DE-DGNs. Therefore, we propose *port-Hamiltonian Deep Graph Network* (PH-DGN) a new message-passing scheme that, by design, introduces the flexibility to balance non-dissipative long-range propagation and non-conservative behaviors as required by the specific task at hand. Therefore, when using purely conservative dynamics, our method allows the preservation and propagation of

---

*Equal Contribution. Correspondence: simon.heilig@rub.de, alessio.gravina@di.unipi.it

long-range information by obeying the conservation laws. In contrast, when our method is used to its full extent, internal damping and additional forces can deviate from this purely conservative behavior, potentially increasing effectiveness in the downstream task. To the best of our knowledge, we are the first to propose a port-Hamiltonian-inspired DE-DGNs. Leveraging the connection with Hamiltonian systems, we provide theoretical guarantees that information is conserved over time. Lastly, the general formulation of our approach can seamlessly incorporate any neighborhood aggregation function (i.e., DGN), thereby endowing these methods with the distinctive properties of our PH-DGN.

Our main contributions can be summarized as follows: (i) We introduce PH-DGN, a novel general DE-DGN inspired by port-Hamiltonian dynamics, which enables the balance and integration of non-dissipative long-range propagation and non-conservative behavior while seamlessly incorporating the most suitable aggregation function; (ii) We theoretically prove that, when pure conservative dynamic is employed, both the continuous and discretized versions of our framework allow for long-range propagation in the message passing flow, since node states retain their past; (iii) We introduce tools inspired by mechanical systems that deviate from such conservative behavior, thus facilitating a clear interpretation from the physics perspective; and (iv) We conduct extensive experiments to demonstrate the benefits of our method and the ability to stack thousands of layers. Our PH-DGN outperforms existing state-of-the-art methods on both synthetic and real-world tasks.

## 2 PORT-HAMILTONIAN DEEP GRAPH NETWORK

We consider the problem of learning node embeddings for a graph $\mathcal{G} = (\mathcal{V}, \mathcal{E})$, where $\mathcal{V}$ is a set of $n$ entities (the nodes) interacting through relations (i.e., edges) in $\mathcal{E} \subseteq \mathcal{V} \times \mathcal{V}$. Each node $u \in \mathcal{V}$ is associated to state $\mathbf{x}_u(t) \in \mathbb{R}^d$, that is the representation of the node at time $t$. The term $\mathbf{X}(t) \in \mathbb{R}^{n \times d}$ is the matrix of all node states in graph $\mathcal{G}$.

We introduce a new DE-DGN framework that designs the information flow within a graph as the solution of a port-Hamiltonian system (Van der Schaft, 2017). Hamiltonian mechanics is a formalism for physical systems based on the Hamiltonian function $H(\mathbf{p}, \mathbf{q}, t)$, which represents the generalized energy of the system with position $\mathbf{q}$ and momentum $\mathbf{p}$. A classic example of a Hamiltonian system is that of a simple mass-spring pendulum, with mass $m$ attached to a spring with constant $k$ having position $q = x$ and momentum $p = m\dot{x}$. The Hamiltonian of the system is the total energy $H = K + P$ where $K$ is the kinetic component $K = \frac{p^2}{2m} = \frac{1}{2}m\dot{x}^2$ and $P$ the spring potential component $P = \frac{1}{2}kx^2$. Hamilton's equations are then defined as:

$$\dot{p} = -\frac{\partial H}{\partial q} = -kx, \quad \dot{q} = \frac{\partial H}{\partial p} = \frac{p}{m}, \tag{1}$$

from which we recover the well-known mass-spring pendulum equation $m\ddot{x} = -kx$. The Hamiltonian formalism allows for an easy description of the dynamics of a system based on its energy and provides theoretical results that will allow us to guarantee relevant properties on our system, such as energy preservation. It is widely used both in classical mechanics (Arnold et al., 2013) as well as in quantum mechanics (Griffiths & Schroeter, 2018) due to its generality. In the port-Hamiltonian formulation, the system allows for energy exchange between subsystems and interaction with external environments. Therefore, port-Hamiltonian systems let us introduce non-conservative phenomena in the system, such as internal dampening $D(q)p$ and external forcing $F(q, t)$, acting on the momentum equation as $\dot{p} = -\frac{\partial H}{\partial q} - D(q)p + F(q, t)$.

Here, we show how the port-Hamiltonian formulation provides the backing to preserve and propagate long-range information between nodes in the absence of non-conservative behaviors, thus in adherence to the laws of conservation. The casting of the system in the more general, full port-Hamiltonian setting, then, introduces the possibility of trading non-dissipation with non-conservative behaviors when needed by the task at hand. Our approach is general, as it can be applied to any message-passing DGN, and frames in a theoretically sound way the integration of non-dissipative propagation and non-conservative behaviors. In the following, we refer to our framework as *port-Hamiltonian Deep Graph Network* (PH-DGN). Figure 1 shows our high-level architecture hinting at how the initial state of the system is propagated up to the terminal time $T$. While the state evolves preserving energy, internal dampening and additional forces (in the following denoted as driving forces) can intervene to alter its conservative trajectory.

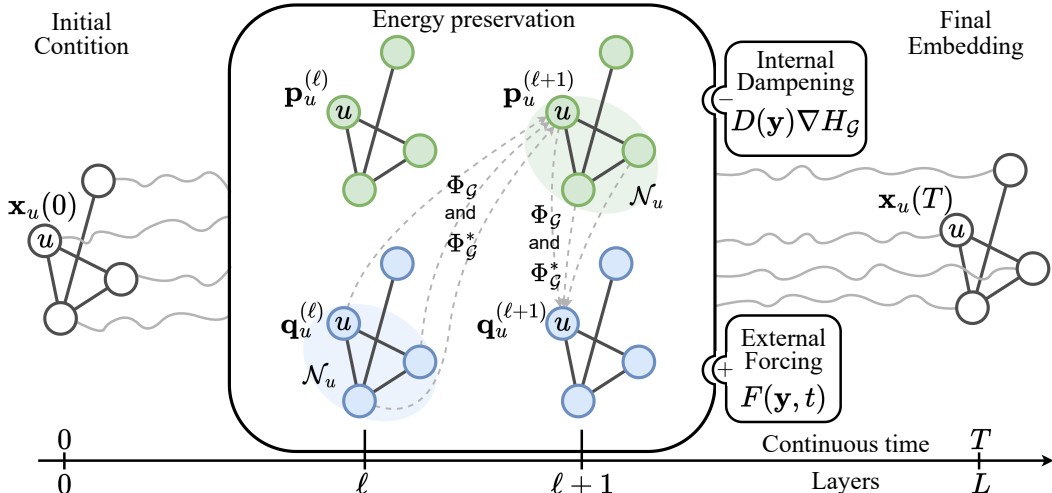

Figure 1: A high-level overview of the proposed port-Hamiltonian Deep Graph Network. It summarizes how the initial node state $\mathbf{x}_u(0)$ is propagated by means of energy preservation up until the terminal time $T$ (i.e., layer $L$), $\mathbf{x}_u(T)$. While the global system's state $\mathbf{y}$ evolves preserving energy, external forces (i.e., dampening $D(\mathbf{y})$ and external control $F(\mathbf{y}, t)$) can intervene to alter its conservative trajectory. The gray trajectories between the initial and final states represent the continuous evolution of the system. The discrete message passing step from layer $\ell$ to $\ell+1$, which is shown in middle of the figure, is given by the coupling of coordinates $\mathbf{q}$ and momenta $\mathbf{p}$ in terms of neighborhood aggregation $\Phi_{\mathcal{G}}$ and influence to adjacent neighbors $\Phi_{\mathcal{G}}^*$. Self-influence on both $\mathbf{q}$ and $\mathbf{p}$ from the previous step $\ell$ are omitted for simplicity.

In the following, we present our method in a bottom-up fashion. Thus, we start by deriving our PH-DGN from a purely conservative system, proving its conservative behavior theoretically, and then extend it by integrating non-conservative behaviors.

**Conservative message passing.** To inject a purely conservative behavior inspired by port-Hamiltoninan dynamics into a DE-DGN, we start by considering the graph Hamiltonian system described by the following ODE

$$\frac{\mathrm{d}\mathbf{y}(t)}{\mathrm{d}t} = \mathcal{J}\nabla H_{\mathcal{G}}(\mathbf{y}(t)), \tag{2}$$

for time $t \in [0, T]$ and subject to an initial condition $\mathbf{y}(0) = \mathbf{y}^0$. The term $\mathbf{y}(t) \in \mathbb{R}^{nd}$ is the vectorized view of $\mathbf{X}(t)$ that represents the global state of the graph at time $t$, with an even dimension $d$, following the notation of Hamiltonian systems (Hairer et al., 2006). $H_{\mathcal{G}} : \mathbb{R}^{nd} \to \mathbb{R}$ is a neural-parameterized Hamiltonian function capturing the energy of the system. The skew-symmetric matrix $\mathcal{J} = \begin{pmatrix} \mathbf{0} & -\mathbf{I}_{nd/2} \\ \mathbf{I}_{nd/2} & \mathbf{0} \end{pmatrix}$, with $\mathbf{I}_{nd/2}$ being the identity matrix of dimension $nd/2$, reflects a rotation of the gradient $\nabla H_{\mathcal{G}}$ and couples the position and momentum of the system.

Since we are dealing with a port-Hamiltonian system, the global state $\mathbf{y}(t)$ is composed by two components which are the momenta, $\mathbf{p}(t) = (\mathbf{p}_1(t), \ldots, \mathbf{p}_n(t))$, and the position, $\mathbf{q}(t) = (\mathbf{q}_1(t), \ldots, \mathbf{q}_n(t))$, of the system, thus $\mathbf{y}(t) = (\mathbf{p}(t), \mathbf{q}(t))$. Therefore, from the node (local) perspective, each node state is expressed as $\mathbf{x}_u(t) = (\mathbf{p}_u(t), \mathbf{q}_u(t))$.

Under this local node-wise perspective, Eq. (2) can be equivalently written as

$$\frac{\mathrm{d}\mathbf{x}_u(t)}{\mathrm{d}t} = \begin{pmatrix} \dot{\mathbf{p}}_u(t) \\ \dot{\mathbf{q}}_u(t) \end{pmatrix} = \begin{pmatrix} -\nabla_{\mathbf{q}_u} H_{\mathcal{G}}\big(\mathbf{p}(t), \mathbf{q}(t)\big) \\ \nabla_{\mathbf{p}_u} H_{\mathcal{G}}\big(\mathbf{p}(t), \mathbf{q}(t)\big) \end{pmatrix}, \ \forall u \in \mathcal{V}. \tag{3}$$

With the aim of designing a purely conservative port-Hamiltonian system (i.e., driving forces are null, reducing it to a Hamiltonian system) based on message passing, we instantiate the Hamiltonian

function $H_{\mathcal{G}}$ as

$$H_{\mathcal{G}}(\mathbf{y}(t)) = \sum_{u \in \mathcal{V}} \tilde{\sigma}(\mathbf{W}\mathbf{x}_u(t) + \Phi_{\mathcal{G}}(\{\mathbf{x}_v(t)\}_{v \in \mathcal{N}_u}) + \mathbf{b})^{\top} \mathbf{1}_d, \quad (4)$$

where $\tilde{\sigma}(\cdot)$ is the anti-derivative of a monotonically non-decreasing activation function $\sigma$, $\mathcal{N}_u$ is the neighborhood of node $u$, and $\Phi_{\mathcal{G}}$ is a neighborhood aggregation permutation-invariant function. Terms $\mathbf{W} \in \mathbb{R}^{d \times d}$ and $\mathbf{b} \in \mathbb{R}^d$ are the weight matrix and the bias vector, respectively, containing the trainable parameters of the system; $\mathbf{1}_d$ denotes a vector of ones of length $d$.

By computing the gradient $\nabla_{\mathbf{x}_u} H_{\mathcal{G}}(\mathbf{y}(t))$ we obtain an explicit version of Eq. (3), which can be rewritten from the node-wise perspective of the information flow as the sum of the self-node evolution influence and its neighbor's evolution influence (referred to as $\Phi_{\mathcal{G}}^*$). More formally, for each node $u \in \mathcal{V}$

$$\frac{\mathrm{d}\mathbf{x}_u(t)}{\mathrm{d}t} = \mathcal{J}_u \Bigg[ \mathbf{W}^{\top} \sigma(\mathbf{W}\mathbf{x}_u(t) + \Phi_{\mathcal{G}}(\{\mathbf{x}_v(t)\}_{v \in \mathcal{N}_u}) + \mathbf{b})$$
$$+ \underbrace{\sum_{v \in \mathcal{N}_u \cup \{u\}} \left(\frac{\partial \Phi_{\mathcal{G}}(\{\mathbf{x}_z(t)\}_{z \in \mathcal{N}_v})}{\partial \mathbf{x}_u(t)}\right)^{\top} \sigma(\mathbf{W}\mathbf{x}_v(t) + \Phi_{\mathcal{G}}(\{\mathbf{x}_z(t)\}_{z \in \mathcal{N}_v}) + \mathbf{b})}_{\Phi_{\mathcal{G}}^*} \Bigg]. \quad (5)$$

Here, $\mathcal{J}_u$ has the same structure as $\mathcal{J}$, but the identity blocks have dimension $d/2$ as it is applied to the single node $u$. Notice that the system in Eq. (5) implements a Hamiltonian system, so it adheres solely to conservation laws.

Now, given an initial condition $\mathbf{x}_u(0)$ for a node $u$, and the other nodes in the graph, the ODE defined in Eq. (5) is a continuous information processing system over a graph governed by conservation laws that computes the final node representation $\mathbf{x}_u(T)$. This is visually summarized in Figure 1 when dampening and external forcing are excluded.

Moreover, we observe that the general formulation of the neighborhood aggregation function $\Phi_{\mathcal{G}}(\{\mathbf{x}_v(t)\}_{v \in \mathcal{N}_u})$ allows implementing any function that aggregates nodes (and edges) information. Therefore, $\Phi_{\mathcal{G}}(\{\mathbf{x}_v(t)\}_{v \in \mathcal{N}_u})$ allows enhancing a standard DGN with our Hamiltonian conservation. As a demonstration of this, in Section 3, we experiment with two neighborhood aggregation functions, which are the classical GCN aggregation (Kipf & Welling, 2017) and

$$\Phi_{\mathcal{G}}(\{\mathbf{x}_v(t)\}_{v \in \mathcal{N}_u}) = \sum_{v \in \mathcal{N}_u} \mathbf{V}\mathbf{x}_v(t). \quad (6)$$

Further details about the discretization of the purely conservative PH-DGN are in Appendix A.3.

**Purely conservative PH-DGN allows long-range propagation.** We show that our PH-DGN in Eq. (5) adheres to the laws of conservation, allowing long-range propagation in the message-passing flow.

As discussed in (Haber & Ruthotto, 2017; Gravina et al., 2023), non-dissipative propagation is directly linked to the sensitivity of the solution of the ODE to its initial condition, thus to the stability of the system. Such sensitivity is controlled by the Jacobian's eigenvalues of Eq. (5). Under the assumption that the Jacobian varies sufficiently slow over time and its eigenvalues are purely imaginary, then the initial condition is effectively propagated into the final node representation, making the system both stable and non-dissipative, thus allowing for long-range propagation.

**Theorem 2.1.** *The Jacobian matrix of the system defined by the ODE in Eq. (5) possesses eigenvalues purely on the imaginary axis, i.e.,*

$$\mathrm{Re}\left(\lambda_i \left(\frac{\partial}{\partial \mathbf{x}_u} \mathcal{J}_u \nabla_{\mathbf{x}_u} H_{\mathcal{G}}(\mathbf{y}(t))\right)\right) = 0, \quad \forall i,$$

*where $\lambda_i$ represents the $i$-th eigenvalue of the Jacobian.*

We report the proof in Appendix B.1. Then, we take a further step and strengthen such result by proving that the nonlinear vector field defined by conservative PH-DGN is divergence-free, thus

preserving information within the graph during the propagation process and helping to maintain informative node representations. In other words, the PH-DGN's dynamics possess a non-dissipative behavior independently of both the assumption regarding the slow variation of the Jacobian and the position of the Jacobian eigenvalues on the complex plane.

**Theorem 2.2.** *The autonomous Hamiltonian $H_{\mathcal{G}}$ of the system in Eq. (5) with learnable weights shared across time stays constant at the energy level specified by the initial value $H_{\mathcal{G}}(\mathbf{y}(0))$, i.e., $\mathrm{d}H_{\mathcal{G}}/\mathrm{d}t = 0$, and possesses a divergence-free nonlinear vector field*

$$\nabla \cdot \mathcal{J}_u \nabla_{\mathbf{x}_u} H_{\mathcal{G}}(\mathbf{y}(t)) = 0, \quad t \in [0, T]. \tag{7}$$

See proof in Appendix B.2. This allows us to interpret the system dynamics as purely rotational, without energy loss, and demonstrates that our PH-DGN is governed by conservation laws when driving forces are null.

We now provide a sensitivity analysis, following Chang et al. (2019); Gravina et al. (2023); Galimberti et al. (2023), to prove that our conservative PH-DGN effectively allows for long-range information propagation. Specifically, we measure the sensitivity of a node state after an arbitrary time $T$ of the information propagation with respect to its previous state, $\|\partial \mathbf{x}_u(T)/\partial \mathbf{x}_u(T-t)\|$. In other words, we compute the backward sensitivity matrix (BSM). We now provide a theoretical bound of our PH-DGN, with its proof in Appendix B.3.

**Theorem 2.3.** *Considering the continuous system defined by Eq. (5), the backward sensitivity matrix (BSM) is bounded from below:*

$$\left\| \frac{\partial \mathbf{x}_u(T)}{\partial \mathbf{x}_u(T-t)} \right\| \geq 1, \quad \forall t \in [0, T].$$

We note that since $\partial \mathbf{x}_u(T)/\partial \mathbf{x}_u(T-t) \in \mathbb{R}^{d \times d}$, our statement is valid for all sub-multiplicative matrix norms, e.g., p-norm and Frobenius norm. The result of Theorem 2.3 indicates that the gradients in the backward pass do not vanish, enabling the effective propagation of previous node states through successive transformations to the final nodes' representations. Therefore, whenever driving forces are null, PH-DGN has a conservative message passing, where the final representation of each node retains its complete past. We observe that Theorem 2.3 holds even during discretization when the Symplectic Euler method is employed (see Appendix A.3).

To give the full picture of the time dynamics of the gradients, we present a similar analysis and provide an upper bound of the BSM in Theorem A.1. Recently, Topping et al. (2022); Di Giovanni et al. (2023) proposed to evaluate the long-range propagation ability of a model by measuring the sensitivity of the node embedding after $\ell$ layers with respect to the input of another node, i.e., $\|\partial \mathbf{x}_u^{(\ell)}/\partial \mathbf{x}_v^{(0)}\|$, bounding such a measure on a MPNN:

$$\left\| \frac{\partial \mathbf{x}_v^{(\ell)}}{\partial \mathbf{x}_u^{(0)}} \right\|_{L_1} \leq (c_\sigma w d)^\ell ((c_r \mathbf{I} + c_a \mathbf{A})^\ell)_{vu} \tag{8}$$

where $c_\sigma$ is the Lipschitz constant of non linearity $\sigma$, $w$ is the maximal entry-value over all weight matrices, $d$ is the embedding dimension, and $c_r$ and $c_a$ being the weighted contributions of the residual and aggregation term, respectively. Following a similar analysis, in Theorem 2.4 we provide a bound for our PH-DGN when Symplectic Euler method is used as discretization method.

**Theorem 2.4.** *Considering our PH-DGN discretized via Symplectic Euler method (Eqs. (12) and (13)), with neighborhood aggregation function of the form $\Phi_{\mathcal{G}} = \sum_{v \in \mathcal{N}_u} \mathbf{V}\mathbf{x}_v$, then*

$$\left\| \frac{\partial \mathbf{x}_u^{(\ell)}}{\partial \mathbf{x}_v^{(0)}} \right\|_{L_1} \leq (dwNc_\sigma)^\ell ((w\mathbf{I} + w(N+1)\mathbf{A})^\ell)_{uv}, \tag{9}$$

*where $N = \max_u |\mathcal{N}_u|$, and $c_r = c_a = 1$ for simplicity.*

See proof in Appendix B.6. The result of Theorem 2.4 indicates that our upper bound on the right-hand side of the inequality is **at least** $N^\ell$ times bigger than the one computed for an MPNN. Therefore, together with previous theoretical results, it holds the capability of PH-DGN to perform long-range propagation effectively.

**Introducing dissipative components.** Without driving forces, a purely conservative inductive bias forces the node states to follow trajectories that maintain constant energy, potentially limiting the effectiveness of the DGN on downstream tasks by restricting the system's ability to model all complex nonlinear dynamics. To this end, we complete the formalization of our port-Hamiltonian framework by introducing tools from mechanical systems, such as friction and external control, to learn how much the dynamic should deviate from this purely conservative behavior. Therefore, we extend the dynamics in Eq. (5) by including two new terms $D(\mathbf{q}) \in \mathbb{R}^{d/2 \times d/2}$ and $F(\mathbf{q}, t) \in \mathbb{R}^{d/2}$, i.e.,

$$\frac{\mathrm{d}\mathbf{x}_u(t)}{\mathrm{d}t} = \left[ \mathcal{J}_u - \begin{pmatrix} D(\mathbf{q}(t)) & \mathbf{0} \\ \mathbf{0} & \mathbf{0} \end{pmatrix} \right] \nabla_{\mathbf{x}_u} H_{\mathcal{G}}(\mathbf{y}(t)) + \begin{pmatrix} F(\mathbf{q}(t), t) \\ \mathbf{0} \end{pmatrix}, \quad \forall u \in \mathcal{V}. \tag{10}$$

Depending on the definition of $D(\mathbf{q}(t))$ we can implement different forces. Specifically, if $D(\mathbf{q}(t))$ is positive semi-definite then it implements internal dampening, while a negative semi-definite implementation leads to internal acceleration. A mixture of dampening and acceleration is obtained otherwise. In the case of dampening, the energy is decreased along the flow of the system (Van der Schaft, 2017). To further enhance the modeling capabilities, we integrate the learnable state- and time-dependent external force $F(\mathbf{q}(t), t)$, which further drives node representation trajectories. Figure 1 visually summarizes how such tools can be plugged in our framework during node update.

Although $D(\mathbf{q}(t))$ and $F(\mathbf{q}(t), t)$ can be implemented as static (fixed) functions, in our experiments in Section 3 we employ neural networks to learn such terms. We provide additional details on the specific architectures in Appendix A.2. We provide further details about the discretization of PH-DGN in Appendix A.3. Additional theoremes supporting the long-range propagation capability of our PH-DGN with driving forces are provided in Appendix B.7.

## 3 EXPERIMENTS

We empirically verify both theoretical claims and practical benefits of our framework on popular graph benchmarks for long-range propagation. First (Section 3.1), we conduct a controlled synthetic test showing non-vanishing gradients even when thousands of layers are used. Afterward (Section 3.2), we run a graph transfer task inspired by Di Giovanni et al. (2023) to assess the efficacy in preserving long-range information between nodes. Then, we assess our framework in popular benchmark tasks requiring the exchange of messages at large distances over the graph, including graph property prediction (Section 3.3) and the long-range graph benchmark (Dwivedi et al., 2022) (Section 3.4). An additional ablation on the impact of the different dissipative components on the Minesweeper (Luo et al., 2024) and graph transfer tasks is reported in Appendix D.4. We compare our performance to state-of-the-art methods, such as MPNN-based models, DE-DGNs (including Hamiltonian-inspired DGNs), higher-order DGNs, and graph transformers. Notice that DE-DGNs represent a direct competitor to our method. Additional details on literature methods are in Appendix C.1. We investigate two neighborhood aggregation functions for our PH-DGN, which are the classical GCN aggregation and that in Eq. (6). Our model is implemented in PyTorch (Paszke et al., 2017) and PyTorch-Geometric (Fey & Lenssen, 2019). We release openly the code implementing our methodology and reproducing our empirical analysis at `https://github.com/simonheilig/porthamiltonian-dgn`. Our experimental results were obtained using NVIDIA A100 GPUs and Intel Xeon Gold 5120 CPUs.

### 3.1 NUMERICAL SIMULATIONS

**Setup.** We empirically verify that our theoretical considerations on the purely conservative PH-DGN (i.e., , the driving forces are null) hold true by an experiment requiring to propagate information within a Carbon-60 molecule graph without training on any specific task, i.e., we perform no gradient update step. While doing so, we measure the energy level captured in $H_{\mathcal{G}}(\mathbf{y}(\ell\epsilon))$ in the forward pass and the sensitivity, $\|\partial \mathbf{x}_u^{(L)} / \partial \mathbf{x}_u^{(\ell)}\|$, from each intermediate layer $\ell = 1, \ldots, L$ in the backward pass. We consider the 2-d position of the atom in the molecule as the input node features, fixed terminal propagation time $T = 10$ with various integration step sizes $\epsilon \in \{0.1, 0.01, 0.001\}$ and $T = 300$ with $\epsilon = 0.3$. Note that the corresponding number of layers is computed as $L = T/\epsilon$, i.e., we use tens to thousands of layers. For the ease of the simulation, we use $\tanh$-nonlinearity, fixed learnable weights that are randomly initialized, and the aggregation function in Eq. (6).

**Results.** In Figure 2a, we show the energy difference $H_{\mathcal{G}}(\mathbf{y}(\ell\epsilon)) - H_{\mathcal{G}}(\mathbf{y}(0))$ for different step sizes. For a fixed time $T$, a smaller step size $\epsilon$ is related to a higher number of stacked layers.

We note that the energy difference oscillates around zero, and the smaller the step size the more accurately the energy is preserved. This supports our intuition of the conservative PH-DGN being a discretization of a divergence-free continuous Hamiltonian dynamic, that allows for non-dissipative forward propagation, as stated in Theorem 2.1 and Theorem 2.2. Even for larger step sizes, energy is neither gained nor lost.

Regarding the backward pass, Figures 2b, 2c assert that the lower bound $\|\partial \mathbf{x}(L)/\partial \mathbf{x}(\ell)\| \geq 1$ stated in Theorem 2.3 and its discrete version in Theorem A.2 leads to non-vanishing gradients. In particular, Figure 2c shows a logarithmic-linear increase of sensitivity with respect to the distance to the final layer, hinting at the exponential upper bound derived in Theorem A.1. This growing behavior can be controlled by regularizing the weight matrices, or by use of normalized aggregation functions, as in GCN (Kipf & Welling, 2017).

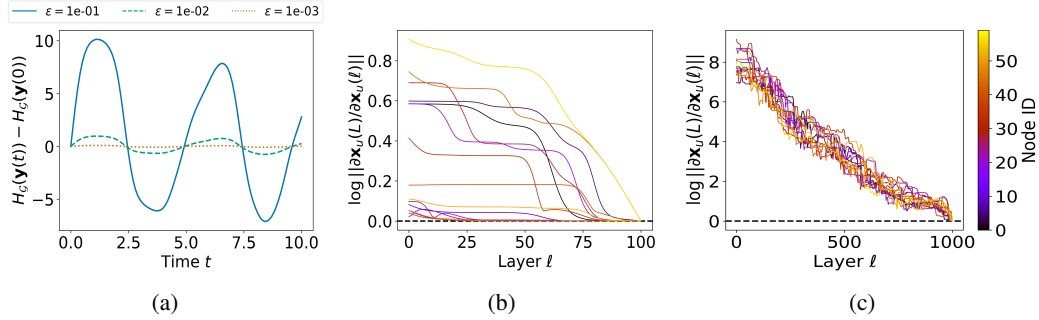

(a)            (b)            (c)

Figure 2: (a) Time evolution of the energy difference to the initial state $\mathbf{y}(0) = \mathbf{y}^0$ obtained from one forward pass of conservative PH-DGN with fixed random weights on the Carbon-60 graph with three different numbers of layers given by $T/\epsilon$. The sensitivity $\|\partial \mathbf{x}_u^{(L)}/\partial \mathbf{x}_u^{(\ell)}\|$ of 15 different node states to their final embedding obtained by backpropagation on the Carbon-60 graph after (b) $T = 10$ and $\epsilon = 0.1$ (i.e., 100 layers) and (c) $T = 300$ and $\epsilon = 0.3$ (i.e., 1000 layers). The log scale's horizontal line at 0 indicates the theoretical lower bound.

## 3.2 GRAPH TRANSFER

**Setup.** We build on the graph transfer tasks by Di Giovanni et al. (2023) and consider the problem of propagating a label from a source node to a target node located at increasing distances $k$ in the graph, following the setting proposed by Gravina et al. (2025a). We use the same graph distributions as in the original work, i.e., line, ring, and crossed-ring graphs. To increase the difficulty of the task, we randomly initialize intermediate nodes with a feature uniformly sampled in $[0, 0.5]$. Source and destination nodes are initialized with labels "1" and "0", respectively. We considered problems at distances $k \in \{3, 5, 10, 50\}$, thus requiring incrementally higher efficacy in propagating long-range information. Given the conservative nature of the task, we focus on assessing our PH-DGN without driving forces. More details on the task and the hyperparameters can be found in the Appendix C.2 and C.5.

**Results.** Figure 3 reports the test mean-squared error (and std. dev.) of PH-DGN compared to literature models. It appears that classical MPNNs do not effectively propagate information across long ranges as their performance decreases when $k$ increases. Differently, PH-DGN achieves low errors even at higher distances, i.e., $k \geq 10$. The only competitor to our PH-DGN is A-DGN, which is another non-dissipative method. Overall, PH-DGN outperforms all the classical MPNNs baseline while having on average better performance than A-DGN, thus empirically supporting our claim of long-range capabilities while introducing a new architectural bias. Moreover, our results highlight how our framework can push simple graph convolutional architectures to state-of-the-art performance when imbuing them with dynamics capable of long-range message exchange.

## 3.3 GRAPH PROPERTY PREDICTION

**Setup.** We consider three graph property prediction tasks introduced in Corso et al. (2020) under the experimental setting of Gravina et al. (2023). We investigate the performance of our port-Hamiltonian

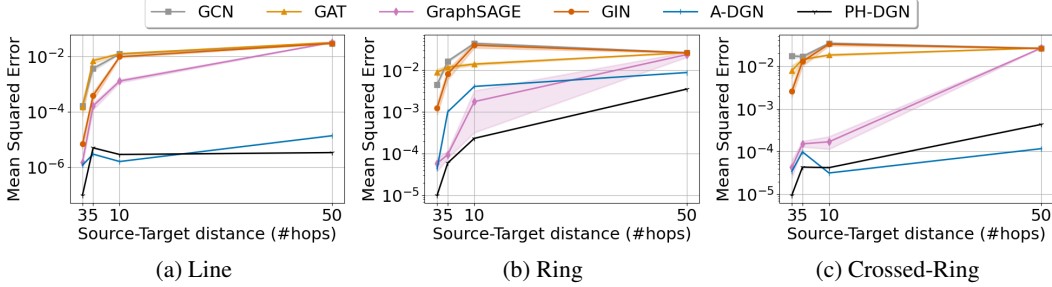

Figure 3: Information transfer performance on (a) Line, (b) Ring, and (c) Crossed-Ring graphs. Overall, baseline approaches are not able to transfer the information accurately as distance increase, while non-dissipative methods like A-DGN and our PH-DGN achieve low errors.

framework in predicting graph diameters, single source shortest paths (SSSP), and node eccentricity on synthetic graphs. Note that to effectively solve these tasks, it is fundamental to propagate not only the information of direct neighborhoods, but also the information coming from nodes far away in the graph. Therefore, good performance in these tasks highlights long-range propagation capabilities. In this experiment, we investigate the performance of our complete framework (i.e., including driving forces), and present the pure conservative PH-DGN (referred to as PH-DGN$_C$) as an ablation study. More details on the task and the hyperparameters can be found in Appendix C.3 and C.5.

**Results.** We present the results on the graph property prediction tasks in Table 1, reporting $log_{10}$(MSE) as evaluation metric. We observe that our PH-DGN show a strong improvement with respect to baseline methods, achieving new state-of-the-art performance on all the tasks. Our ablation reveals that the purely conservative model has, on average, a $log_{10}$(MSE) that is 0.33 lower than the best baseline. Such gap is pushed to 0.81 when the full port-Hamiltonian bias (i.e., PH-DGN) is employed, marking a significant decrease in the test loss. The largest gap is achieved by PH-DGN in the eccentricity task, where it improves the $log_{10}$(MSE) performance of the best baseline by 1.36. Moreover, we note that our PH-DGN is more effective than its purely conservative version and existing Hamiltonian-inspired DE-DGN, i.e., HamGNN and HANG, highlighting the significance of port-Hamiltonian dynamics with respect to a purely conservative inductive bias. This effectiveness is also reflected in the computational cost, as shown in

Table 1: Mean test $log_{10}$(MSE) and std average over 4 training seeds on the Graph Property Prediction. Our methods and DE-DGN baselines are implemented with weight sharing. The **first**, **second**, and **third** best scores are colored.

| Model | Eccentricity | Diameter | SSSP |
|---|---|---|---|
| **MPNNs** | | | |
| GCN | $0.8468_{\pm0.0028}$ | $0.7424_{\pm0.0466}$ | $0.9499_{\pm0.0001}$ |
| GAT | $0.7909_{\pm0.0222}$ | $0.8221_{\pm0.0752}$ | $0.6951_{\pm0.1499}$ |
| GraphSAGE | $0.7863_{\pm0.0207}$ | $0.8645_{\pm0.0401}$ | $0.2863_{\pm0.1843}$ |
| GIN | $0.9504_{\pm0.0007}$ | $0.6131_{\pm0.0990}$ | $-0.5408_{\pm0.4193}$ |
| GCNII | $0.7640_{\pm0.0355}$ | $0.5287_{\pm0.0570}$ | $-1.1329_{\pm0.0135}$ |
| **DE-DGNs** | | | |
| DGC | $0.8261_{\pm0.0032}$ | $0.6028_{\pm0.0050}$ | $-0.1483_{\pm0.0231}$ |
| GraphCON | $0.6833_{\pm0.0074}$ | $0.0964_{\pm0.0620}$ | $-1.3836_{\pm0.0092}$ |
| GRAND | $0.6602_{\pm0.1393}$ | $0.6715_{\pm0.0490}$ | $-0.0942_{\pm0.3897}$ |
| A-DGN | $\mathbf{0.4296}_{\pm0.1003}$ | $\mathbf{-0.5188}_{\pm0.1812}$ | $-3.2417_{\pm0.0751}$ |
| HamGNN | $0.7851_{\pm0.0140}$ | $0.6762_{\pm0.1317}$ | $0.9449_{\pm0.0008}$ |
| HANG | $0.8302_{\pm0.0051}$ | $1.1036_{\pm0.1025}$ | $0.1671_{\pm0.0160}$ |
| **Ours** | | | |
| PH-DGN$_C$ | $-0.7248_{\pm0.1068}$ | $-0.5473_{\pm0.1074}$ | $-3.0467_{\pm0.1615}$ |
| PH-DGN | $-0.9348_{\pm0.2097}$ | $\mathbf{-0.5385}_{\pm0.0187}$ | $-4.2993_{\pm0.0721}$ |

Appendix D.3, where our PH-DGN results to be more efficient both in terms of speed and memory usage compared to HamGNN and HANG.

Although our purely conservative PH-DGN$_C$ shows improved performance with respect to all baselines, it appears that relaxing such bias via PH-DGN is more beneficial overall, leading to even greater improvements in long-range information propagation. Our intuition is that such tasks do not require purely conservative behavior since nodes need to count distances while exchanging more messages with other nodes, similar to standard algorithmic solutions such as Dijkstra (1959). Therefore, the

energy may not be constant during the resolution of the task, hence benefiting from the non-purely conservative behavior of PH-DGN.

As for the graph transfer task, our results demonstrate that our PH-DGNs can effectively learn and exploit long-range information while pushing simple graph neural architectures to state-of-the-art performance when modeling dynamics capable of long-range propagation.

## 3.4 LONG-RANGE GRAPH BENCHMARK

**Setup.** We assess the performance of our method on the real-world long-range graph benchmark (LRGB) from Dwivedi et al. (2022), focusing on the Peptide-func and Peptide-struct tasks. As in Section 3.3, we decouple our method into PH-DGN and PH-DGN$_C$ to provide an ablation study on the strictly conservative behavior. For our evaluation, we follow the experimental setting in Dwivedi et al. (2022). Acknowledging the results from Tönshoff et al. (2023), we also report results with a 3-layer MLP readout. While some baselines leverage positional or structural encodings, our approach does not depend on these mechanisms. More details on the task and the hyperparameters can be found in Appendix C.4 and C.5.

**Results.** We report results on the LRGB tasks in Table 2 (extended results are reported in Appendix D.2 due to space limits). Our results show that both PH-DGN$_C$ and PH-DGN outperform classical MPNNs, graph transformers, most of the multi-hop DGNs, and recent DE-DGNs (which represent a direct competitor to our method). Overall, our port-Hamiltonian framework (with and without driving forces) shows great benefit in propagating long-range information without requiring additional strategies such as global position encoding (as evidenced by comparisons with MPNN-based models using positional encoding), global attention mechanisms (as seen in comparisons with Transformer-based models), or rewiring techniques (as shown in comparisons with Multi-hop DGNs) that increase the overall complexity of the method. Consequently, our results reaffirm the effectiveness of our framework in enabling efficient long-range propagation, even in simple DGNs characterized by purely local message exchanges. Lastly, we believe that our PH-DGN with driving forces achieves better performance than PH-DGN$_C$ on the real-world LRGB because the learned driving forces act as an adaptive filter mechanism that filters noisy information, facilitating the learning of relevant information.

Table 2: Results for Peptides-func and Peptides-struct averaged over 3 training seeds. The **first**, **second**, and **third** best scores are colored. Extended version of this table is provided in Appendix D.2. "+PE/SE" indicates the use of positional or structural encoding. We have detailed the type of encoding wherever the original source explicitly specifies it.

| Model | Peptides-func AP ↑ | Peptides-struct MAE ↓ |
|---|---|---|
| **Modified MPNNs, Tönshoff et al. (2023)** | | |
| GCN+PE/SE | $0.6860_{\pm 0.0050}$ | $0.2460_{\pm 0.0007}$ |
| GCNII+PE/SE | $0.6444_{\pm 0.0011}$ | $0.2507_{\pm 0.0012}$ |
| GINE+PE/SE | $0.6621_{\pm 0.0067}$ | $0.2473_{\pm 0.0017}$ |
| GatedGCN+PE/SE | $0.6765_{\pm 0.0047}$ | $0.2477_{\pm 0.0009}$ |
| **Multi-hop DGNs, Gutteridge et al. (2023)** | | |
| DIGL+MPNN+LapPE | $0.6830_{\pm 0.0026}$ | $0.2616_{\pm 0.0018}$ |
| MixHop-GCN+LapPE | $0.6843_{\pm 0.0049}$ | $0.2614_{\pm 0.0023}$ |
| DRew-GCN+LapPE | $0.7150_{\pm 0.0044}$ | $0.2536_{\pm 0.0015}$ |
| **Transformers, Gutteridge et al. (2023)** | | |
| Transformer+LapPE | $0.6326_{\pm 0.0126}$ | $0.2529_{\pm 0.0016}$ |
| SAN+LapPE | $0.6384_{\pm 0.0121}$ | $0.2683_{\pm 0.0043}$ |
| GraphGPS+LapPE | $0.6535_{\pm 0.0041}$ | $0.2500_{\pm 0.0005}$ |
| **DE-DGNs** | | |
| GRAND | $0.5789_{\pm 0.0062}$ | $0.3418_{\pm 0.0015}$ |
| GraphCON | $0.6022_{\pm 0.0068}$ | $0.2778_{\pm 0.0018}$ |
| A-DGN | $0.5975_{\pm 0.0044}$ | $0.2874_{\pm 0.0021}$ |
| SWAN | $0.6751_{\pm 0.0039}$ | $0.2485_{\pm 0.0009}$ |
| **Ours** | | |
| PH-DGN$_C$ | $0.6961_{\pm 0.0070}$ | $0.2581_{\pm 0.0020}$ |
| PH-DGN | $0.7012_{\pm 0.0045}$ | $0.2465_{\pm 0.0020}$ |

## 4 RELATED WORKS

**DGN based on differential equations.** Recent advancements in the field of representation learning have introduced new architectures that establish a connection between neural networks and dynamical systems. Inspired by pioneering works on recurrent neural networks (Chen et al., 2018; Haber & Ruthotto, 2017; Chang et al., 2019; Galimberti et al., 2023), such connection has been

pushed to the domains of DGNs (Han et al., 2024). Indeed, works like GDE (Poli et al., 2019), GRAND (Chamberlain et al., 2021a), PDE-GCN (Eliasof et al., 2021), DGC (Wang et al., 2021), GRAND++ (Thorpe et al., 2022) propose to interpret DGNs as discretization of ODEs and PDEs. The conjoint of dynamical systems and DGNs have found favorable consensus, as these new methods exploit the intrinsic properties of differential equations to extend the characteristic of message passing within DGNs. GRAND, GRAND++, and DGC bias the node representation trajectories to follow the heat diffusion process, thus performing a gradual smoothing of the initial node states. On the contrary, GraphCON (Rusch et al., 2022) used oscillatory properties to enable linear dynamics that preserve the Dirichlet energy encoded in the node features; PDE-GCN$_M$ (Eliasof et al., 2021) uses an interpolation between anisotropic diffusion and conservative oscillatory properties; more recently, A-DGN (Gravina et al., 2023) introduces an anti-symmetric mechanism that leads to node-wise non-dissipative dynamics. Related work in the line of Hamiltonian systems for DGNs, such as HamGNN (Kang et al., 2023), exclusively leverage Hamiltonian dynamics to encode node input features, which are then fed into classical DGNs to enhance their conservative properties. Similarly, HANG (Zhao et al., 2023) leverages Hamiltonian dynamics to improve robustness to adversarial attacks to the graph structure. Differently from HamGNN and HANG, our PH-DGN is (to the best of our knowledge) the first DGN that leverages port-Hamiltonian dynamics, thus balancing non-dissipative and non-conservative behaviors while providing theoretical guarantees of long-range propagation. A deeper discussion on the differences with HamGNN and HANG is provided in Appendix D.3.

While the aforementioned works focus on the spatial aggregation term of DE-DGNs, the temporal domain has also been studied in works such as Eliasof et al. (2024); Gravina et al. (2024a); Jin et al. (2022); Gravina et al. (2024b).

**Long-range propagation on DGNs.** Effectively transferring information across distant nodes is still an open challenge in the graph representation learning community (Shi et al., 2023). Various strategies have been explored in recent years to address this challenge, including regularizing the model's weight space (Gravina et al., 2023; 2025b;a), filter messages in the information flow (Errica et al., 2024), and graph rewiring. In this latter setting, methods like SDRF (Topping et al., 2022), GRAND (Chamberlain et al., 2021a), BLEND (Chamberlain et al., 2021b), and DRew (Gutteridge et al., 2023) (dynamically) alter the original edge set to densify the graph during preprocessing to facilitate node communication. Differently, Transformer-based methods (Shi et al., 2021; Dwivedi & Bresson, 2021; Ying et al., 2021; Wu et al., 2023) enable message passing between all node pairs. FLODE (Maskey et al., 2024) incorporates non-local dynamics by using a fractional power of the graph shift operator. Although these techniques are effective in addressing the problem of long-range communication, they can also increase the complexity of information propagation due to denser graph shift operators.

## 5 CONCLUSIONS

In this paper, we have presented *port-Hamiltonian Deep Graph Network* (PH-DGN), a general framework that gauges the equilibrium between non-dissipative long-range propagation and non-conservative behavior while seamlessly incorporating the most suitable neighborhood aggregation function. We theoretically prove that, when pure conservative dynamic is employed, both the continuous and discretized versions of our framework allow for long-range propagation in the message passing flow since node states retain their past. To demonstrate the benefits of including port-Hamiltonian dynamics in DE-DGNs, we conducted several experiments on synthetic and real-world benchmarks requiring long-range interaction. Our results show that our method outperforms state-of-the-art models and that the inclusion of data-driven forces that deviate from a purely conservative behavior is often key to maximize efficacy of the approach on tasks requiring long-range propagation. Indeed, in practice, effective information propagation requires a balance between long-term memorization and propagation and the ability to selectively discard and forget information when necessary. Looking ahead to future developments, our port-Hamiltonian dynamic can be extended to handle time-varying streams of graphs (Gravina & Bacciu, 2024) and can be evaluated with alternative discretization methods, e.g., adaptive multistep schemes (Rufai et al., 2023).

## ETHICS AND REPRODUCIBILITY STATEMENTS

**Ethics Statement.** In this work, we do not release any datasets or models that could pose a significant risk of misuse. We believe our research does not have any direct or indirect negative societal implications or harmful consequences, as we do not utilize sensitive, privacy-related data, nor do we develop methods that could be applied for harmful purposes. As far as we are aware, this study does not raise any ethical concerns or potential negative impacts. Furthermore, our research does not involve human subjects, nor does it employ crowdsourcing methods. We confirm there are no potential conflicts of interest or sponsorship affecting the objectivity or outcomes of this study.

**Reproducibility Statement.** In Section 3 we outline the setups employed in our experiments, while in Appendix C we provide comprehensive supplementary information, including references to the baselines, detailed dataset descriptions, the experimental settings for each task, and the hyperparameter grids used in our study. All experiments presented in Sections 3.3 and 3.4 are conducted on publicly available benchmarks. To further support reproducibility, we openly release all the data and code to reproduce our empirical evaluation upon acceptance.

### ACKNOWLEDGMENTS

This research was partially supported by EU-EIC EMERGE (Grant No. 101070918), as well as partially funded in the course of TRR 391 Spatio-temporal Statistics for the Transition of Energy and Transport (520388526) by the Deutsche Forschungsgemeinschaft (DFG, German Research Foundation). SH acknowledges funding from ELSA – European Lighthouse on Secure and Safe AI (Grant No. 101070617) by the European Union.

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

# A ADDITIONAL DETAILS OF THE (PORT-)HAMILTONIAN FRAMEWORK

## A.1 SENSITIVITY UPPER BOUND

Although the sensitivity of a node state after a time $t$ with respect to its previous state can be bounded from below, allowing effective conservative message passing in PH-DGN, we observe that it is possible to compute an upper bound on such a measure, which we provide in the following theorem. While the theorem shows that, theoretically, the sensitivity measure may grow (i.e., potentially causing gradient explosion), we emphasize that during our experiments we did not encounter such a problem.

**Theorem A.1.** *Consider the continuous system defined by Eq. (5), if $\sigma$ is a non-linear function with bounded derivative, i.e. $\exists M > 0, |\sigma'(x)| \leq M$, and the neighborhood aggregation function is of the form $\Phi_{\mathcal{G}} = \sum_{v \in \mathcal{N}_u} \mathbf{V}\mathbf{x}_v$, the backward sensitivity matrix (BSM) is bounded from above:*

$$\left\| \frac{\partial \mathbf{x}_u(T)}{\partial \mathbf{x}_u(T-t)} \right\| \leq \sqrt{d}\, exp(QT), \quad \forall t \in [0, T],$$

*where $Q = \sqrt{d}\, M\|\mathbf{W}\|_2^2 + \sqrt{d}\, Mmax_{i \in [n]}|\mathcal{N}_i|\|\mathbf{V}\|_2^2$.*

We give the proof of this theorem in Appendix B.4.

## A.2 ARCHITECTURAL DETAILS OF DISSIPATIVE COMPONENTS

As typically employed, we follow physics-informed approaches that learn how much dissipation and external control is necessary to model the observations (Desai et al., 2021). In particular, we consider these (graph-) neural network architectures for the dampening term $D(\mathbf{q})$ and external force term $F(\mathbf{q}, t)$, assuming for simplicity $\mathbf{q}_u \in \mathbb{R}^{\frac{d}{2}}$.

**Dampening** $D(\mathbf{q})$: it is a square $\frac{d}{2} \times \frac{d}{2}$ matrix block with only diagonal entries being non-zero and defined as:

- *param*: a learnable vector $\mathbf{w} \in \mathbb{R}^{\frac{d}{2}}$.

- *param+*: a learnable vector followed by a ReLU activation, i.e., $\text{ReLU}(\mathbf{w}) \in \mathbb{R}^{\frac{d}{2}}$.

- *MLP4-ReLU*: a 4-layer MLP with ReLU activation and all layers of dimension $\frac{d}{2}$.

- *DGN-ReLU*: a DGN node-wise aggregation layer from Eq. (6) with ReLU activation.

**External forcing** $F(\mathbf{q}, t)$: it is a $\frac{d}{2}$ dimensional vector where each component is the force on a component of the system. Since it takes as input $\frac{d}{2} + 1$ components it is defined as:

- *MLP4-Sin*: 3 linear layers of $\frac{d}{2} + 1$ units with sin activation followed by a last layer with $\frac{d}{2}$ units.

- *DGN-tanh*: a single node-wise DGN aggregation from Eq. (6) followed by a tanh activation.

Note that dampening, i.e., energy loss, is only given when $D(\mathbf{q})$ represents a positive semi-definite matrix. Hence, we used ReLU-activation, except for *param*, which offers a flexible trade-off between dampening and acceleration learned by backpropagation.

## A.3 Discretization of port-Hamiltonian DGNs

As for standard DE-DGNs a numerical discretization method is needed to solve Eq. (5). However, as observed in Haber & Ruthotto (2017); Galimberti et al. (2023), not all standard techniques can be employed for solving Hamiltonian systems. Indeed, symplectic integration methods need to be used to preserve the conservative properties in the discretized system.

For the ease of simplicity, in the following we focus on the Symplectic Euler method, however, we observe that more complex methods such as Strömer-Verlet can be employed (Hairer et al., 2006).

The Symplectic Euler scheme, applied to our PH-DGN with null driving forces in Eq. (5), updates the node representation at the $(\ell + 1)$-th step as

$$\mathbf{x}_u^{(\ell+1)} = \begin{pmatrix} \mathbf{p}_u^{(\ell+1)} \\ \mathbf{q}_u^{(\ell+1)} \end{pmatrix} = \begin{pmatrix} \mathbf{p}_u^{(\ell)} \\ \mathbf{q}_u^{(\ell)} \end{pmatrix} + \epsilon \mathcal{J}_u \begin{pmatrix} \nabla_{\mathbf{p}_u} H_{\mathcal{G}}(\mathbf{p}^{(\ell+1)}, \mathbf{q}^{(\ell)}) \\ \nabla_{\mathbf{q}_u} H_{\mathcal{G}}(\mathbf{p}^{(\ell+1)}, \mathbf{q}^{(\ell)}) \end{pmatrix}, \quad \forall u \in \mathcal{V}. \tag{11}$$

with $\epsilon$ the step size of the numerical discretization. We note that Eq. (11) relies on both the current and future state of the system, hence marking an implicit scheme that would require solving a linear system of equations in each step. To obtain an explicit version of Eq. (11), we consider the neighborhood aggregation function in Eq. (6) and impose a structure assumption on $\mathbf{W}$ and $\mathbf{V}$, namely $\mathbf{W} = \begin{pmatrix} \mathbf{W}_p & \mathbf{0} \\ \mathbf{0} & \mathbf{W}_q \end{pmatrix}$ and $\mathbf{V} = \begin{pmatrix} \mathbf{V}_p & \mathbf{0} \\ \mathbf{0} & \mathbf{V}_q \end{pmatrix}$. A comparable assumption can be made for other neighborhood aggregation functions, such as GCN aggregation. Despite the necessary block diagonal structure assumption on $\mathbf{W}$ and $\mathbf{V}$ to ensure the separation into the $\mathbf{p}$ and $\mathbf{q}$ components, we note that $\mathbf{W}_p$, $\mathbf{W}_q$, $\mathbf{V}_p$, and $\mathbf{V}_q$ are unconstrained learnable weight matrices.

Therefore, the gradients in Eq. (11) can be rewritten in the explicit form as

$$\mathbf{p}_u^{(\ell+1)} = \mathbf{p}_u^{(\ell)} - \epsilon\left[\mathbf{W}_q^\top \sigma(\mathbf{W}_q\mathbf{q}_u^{(\ell)} + \Phi_{\mathcal{G}}(\{\mathbf{q}_v^{(\ell)}\}_{v\in\mathcal{N}_u}) + \mathbf{b}_q)\right.$$

$$\left. + \sum_{v\in\mathcal{N}_u\setminus\{u\}} \mathbf{V}_q^\top \sigma(\mathbf{W}_q\mathbf{q}_v^{(\ell)} + \Phi_{\mathcal{G}}(\{\mathbf{q}_j^{(\ell)}\}_{j\in\mathcal{N}_v}) + \mathbf{b}_q)\right] \tag{12}$$

$$\mathbf{q}_u^{(\ell+1)} = \mathbf{q}_u^{(\ell)} + \epsilon\left[\mathbf{W}_p^\top \sigma(\mathbf{W}_p\mathbf{p}_u^{(\ell+1)} + \Phi_{\mathcal{G}}(\{\mathbf{p}_v^{(\ell+1)}\}_{v\in\mathcal{N}_u}) + \mathbf{b}_p)\right.$$

$$\left. + \sum_{v\in\mathcal{N}_u\setminus\{u\}} \mathbf{V}_p^\top \sigma(\mathbf{W}_p\mathbf{p}_v^{(\ell+1)} + \Phi_{\mathcal{G}}(\{\mathbf{p}_j^{(\ell+1)}\}_{j\in\mathcal{N}_v}) + \mathbf{b}_p)\right]. \tag{13}$$

We observe that Eqs. (12) and (13) can be understood as coupling two DGN layers. This discretization mechanism is visually summarized in the middle of Figure 1 where a message-passing step from layer $\ell$ to layer $\ell+1$ is performed.

In the case of PH-DGN with driving forces in Eq. (10) the discretization employs the same step for $\mathbf{q}^{(\ell+1)}$ in Eq. (13) while Eq. (12) includes the dissipative components, thus it can be rewritten as

$$\mathbf{p}_u^{(\ell+1)} = \mathbf{p}_u^{(\ell)} + \epsilon\left[-\nabla_{\mathbf{q}_u}H_{\mathcal{G}}(\mathbf{p}^{(\ell)}, \mathbf{q}^{(\ell)}) - D_u(\mathbf{q}^{(\ell+1)})\nabla_{\mathbf{p}_u}H_{\mathcal{G}}(\mathbf{p}^{(\ell)}, \mathbf{q}^{(\ell)}) + F_u(\mathbf{q}^{(\ell)}, t)\right]. \tag{14}$$

Lastly, it is important to acknowledge that properties observed in the continuous domain may not necessarily hold in the discrete setting due to the limitations of the discretization method. In the following theorem, we show that when the Symplectic Euler method is employed, then Theorem 2.3 holds.

**Theorem A.2.** *Considering the discretized system in Eq. (11) obtained by Symplectic Euler discretization, the backward sensitivity matrix (BSM) is bounded from below:*

$$\left\|\frac{\partial\mathbf{x}_u^{(L)}}{\partial\mathbf{x}_u^{(L-\ell)}}\right\| \geq 1, \quad \forall\ell\in[0,L].$$

We provide the proof in Appendix B.5. Again, this indicates that even the discretized version of PH-DGN with null driving forces enables for effective propagation and conservative message passing.

## B  PROOFS OF THE THEORETICAL STATEMENTS

In this section, we provide the proofs of the theoretical statements in the main text and in appendix A. As for the rest of the paper, we will use the denominator notation, i.e., Jacobian matrices have output components on columns and input components on the rows.

### B.1  PROOF OF THEOREM 2.1

*Proof.* First, we note that

$$\frac{\partial}{\partial\mathbf{x}_u}\mathcal{J}_u\nabla_{\mathbf{x}_u}H_{\mathcal{G}}(\mathbf{y}(t)) = \nabla_{\mathbf{x}_u}^2 H_{\mathcal{G}}(\mathbf{y}(t))\mathcal{J}_u^\top, \tag{15}$$

where $\nabla_{\mathbf{x}_u}^2 H_{\mathcal{G}}$ is the symmetric Hessian matrix. Hence, the Jacobian is shortly written as $\mathbf{AB}$, where $\mathbf{A}$ is symmetric and $\mathbf{B}$ is anti-symmetric. Consider an eigenpair of $\mathbf{AB}$, where the eigenvector is denoted by $\mathbf{v}$ and the eigenvalue by $\lambda\neq 0$. Then:

$$\mathbf{v}^*\mathbf{AB} = \lambda\mathbf{v}^*$$

$$\mathbf{v}^*\mathbf{A} = \lambda\mathbf{v}^*\mathbf{B}^{-1}$$

$$\mathbf{v}^*\mathbf{Av} = \lambda\left(\mathbf{v}^*\mathbf{B}^{-1}\mathbf{v}\right)$$

where $*$ represents the conjugate transpose. On the left-hand side, it is noticed that the $(\mathbf{v}^* \mathbf{A} \mathbf{v})$ term is a real number. Recalling that $\mathbf{B}^{-1}$ remains anti-symmetric and for any real anti-symmetric matrix $\mathbf{C}$ it holds that $\mathbf{C}^* = \mathbf{C}^\top = -\mathbf{C}$, it follows that $(\mathbf{v}^* \mathbf{C} \mathbf{v})^* = \mathbf{v}^* \mathbf{C}^* \mathbf{v} = -\mathbf{v}^* \mathbf{C} \mathbf{v}$. Hence, the $\mathbf{v}^* \mathbf{B}^{-1} \mathbf{v}$ term on the right-hand side is an imaginary number. Thereby, $\lambda$ needs to be purely imaginary, and, as a result, all eigenvalues of $\mathbf{A} \mathbf{B}$ are purely imaginary. $\qquad \square$

## B.2 PROOF OF THEOREM 2.2

Our conservative PH-DGN has shared weights $\mathbf{W}, \mathbf{V}$ across the layers of the DGN. This means that the Hamiltonian is autonomous and does not depend explicitly on time $H_{\mathcal{G}}(\mathbf{y}(t), t) = H_{\mathcal{G}}(\mathbf{y}(t))$ as we can see from Eq. (4). In such case, the energy is naturally conserved in the system it represents: this is a consequence of the Hamiltonian flow and dynamics, as we show in this theorem. For a more in-depth description and analysis of Hamiltonian dynamics in general we refer to Arnold et al. (2013).

*Proof.* The time derivative of $H(\mathbf{y}(t))$ is given by means of the chain-rule:

$$\frac{\mathrm{d}H(\mathbf{y}(t))}{\mathrm{d}t} = \frac{\partial H(\mathbf{y}(t))}{\partial \mathbf{y}(t)} \cdot \frac{\mathrm{d}\mathbf{y}(t)}{\mathrm{d}t} = \frac{\partial H(\mathbf{y}(t))}{\partial \mathbf{y}(t)} \cdot \mathcal{J} \frac{\partial H(\mathbf{y}(t))}{\partial \mathbf{y}(t)} = 0, \tag{16}$$

where the last equality holds since $\mathcal{J}$ is anti-symmetric. Having no change over time implies that $H(\mathbf{y}(t)) = H(\mathbf{y}(0)) = \text{const}$ for all $t$.

Since the Hessian $\nabla^2 H_{\mathcal{G}}(\mathbf{y}(t))$ is symmetric, it follows directly

$$\nabla \cdot \mathcal{J}_u \nabla_{\mathbf{x}_v} H_{\mathcal{G}}(\mathbf{y}(t)) = \sum_{i=1}^{d} -\frac{\partial^2 H_{\mathcal{G}}(\mathbf{y}(t))}{\partial q_v^i \, \partial p_v^i} + \frac{\partial^2 H_{\mathcal{G}}(\mathbf{y}(t))}{\partial p_v^i \, \partial q_v^i} = 0$$

$\qquad \square$

## B.3 PROOF OF THEOREM 2.3

In order to prove the lower bound on the BSM, we need a technical lemma that describes the time evolution of the BSM itself, which extends the result from Galimberti et al. (2023).

**Lemma B.1.** *Given the system dynamics of the ODE in Eq. (2), we have that*

$$\frac{\mathrm{d}}{\mathrm{d}t} \frac{\partial \mathbf{y}(T)}{\partial \mathbf{y}(T-t)} = \mathcal{J} \left. \frac{\partial H}{\partial \mathbf{y}} \right|_{\mathbf{y}(T-t)} \frac{\partial \mathbf{y}(T)}{\partial \mathbf{y}(T-t)} \tag{17}$$

*as in (Galimberti et al., 2023). The same applies, with a slightly different formula, for each node $u$, that is the BSM satisfies*

$$\frac{\mathrm{d}}{\mathrm{d}t} \frac{\partial \mathbf{x}_u(T)}{\partial \mathbf{x}_u(T-t)} = \left. \frac{\partial \mathbf{y}}{\partial \mathbf{x}_u} \right|_{(T-t)} \left. \frac{\partial f_u}{\partial \mathbf{y}} \right|_{\mathbf{y}(T-t)} \frac{\partial \mathbf{x}_u(T)}{\partial \mathbf{x}_u(T-t)} = F_u \frac{\partial \mathbf{x}_u(T)}{\partial \mathbf{x}_u(T-t)} \tag{18}$$

*where $f_u$ is the restriction of $f = \mathcal{J}\frac{\partial H}{\partial \mathbf{y}}$ to the components corresponding to $\mathbf{x}_u$, that is the dynamics of node $u$, which can be written as*

$$f_u = L_u f = L_u \mathcal{J} \frac{\partial H}{\partial \mathbf{y}} \tag{19}$$

*where $L_u$ is the readout matrix, of the form*

$$L_u = \begin{bmatrix} 0_{\frac{d}{2} \times \frac{d}{2}(u-1)} & I_{\frac{d}{2} \times \frac{d}{2}} & 0_{\frac{d}{2} \times \frac{d}{2}(n-u)} & 0_{\frac{d}{2} \times \frac{d}{2}(u-1)} & 0_{\frac{d}{2} \times \frac{d}{2}} & 0_{\frac{d}{2} \times \frac{d}{2}(n-u)} \\ 0_{\frac{d}{2} \times \frac{d}{2}(u-1)} & 0_{\frac{d}{2} \times \frac{d}{2}} & 0_{\frac{d}{2} \times \frac{d}{2}(n-u)} & 0_{\frac{d}{2} \times \frac{d}{2}(u-1)} & I_{\frac{d}{2} \times \frac{d}{2}} & 0_{\frac{d}{2} \times \frac{d}{2}(n-u)} \end{bmatrix} \tag{20}$$

*which is a projection on the coordinates of a single node $u$. Notice as well that, in denominator notation*

$$\frac{\partial \mathbf{y}}{\partial \mathbf{x}_u} = L_u \tag{21}$$

We only show Eq. (18), as it is related to graph networks and is actually a harder version of Eq. (17), with the latter being already proven in Galimberti et al. (2023). We also show the last part of the proof in a general sense, without using the specific matrices of the Hamiltonian used.

*Proof.* Following Galimberti et al. (2023), the solution to the ODE $\frac{\mathrm{d}\mathbf{x}_u}{\mathrm{d}t} = f_u(\mathbf{y}(t))$ can be written in integral form as

$$\mathbf{x}_u(T) = \mathbf{x}_u(T-t) + \int_{T-t}^{T} f_u(\mathbf{y}(\tau))\mathrm{d}\tau = \mathbf{x}_u(T-t) + \int_0^t f_u(\mathbf{y}(T-t+s))\mathrm{d}s \quad (22)$$

Differentiating by the solution at a previous time $\mathbf{x}_u(T-t)$ we obtain

$$\begin{aligned}
\frac{\partial \mathbf{x}_u(T)}{\partial \mathbf{x}_u(T-t)} &= I_u + \frac{\partial \int_0^{\top} f_u(\mathbf{y}(T-t+s))\mathrm{d}s}{\partial \mathbf{x}_u(T-t)} = \\
&= I_u + \int_0^t \frac{\partial f_u(\mathbf{y}(T-t+s))}{\partial \mathbf{x}_u(T-t)} \\
&= I_u + \int_0^t \frac{\partial \mathbf{y}(T-t+s)}{\partial \mathbf{x}_u(T-t)} \left.\frac{\partial f_u}{\partial \mathbf{y}}\right|_{\mathbf{y}(T-t+s)} \mathrm{d}s
\end{aligned} \quad (23)$$

where in the second equality we brought the derivative term under the integral sign and in the third we used the chain rule of the derivative (recall we are using denominator notation). Considering a slight perturbation in time $\delta$, we consider $\frac{\partial \mathbf{x}_u(T)}{\partial \mathbf{x}_u(T-t-\delta)}$ as this will be used to calculate the time derivative of the BSM. Using again the chain rule for the derivative and the formula above with $T-t-\delta$ instead of $T-t$, we have that

$$\begin{aligned}
\frac{\partial \mathbf{x}_u(T)}{\partial \mathbf{x}_u(T-t-\delta)} &= \frac{\partial \mathbf{x}_u(T-t)}{\partial \mathbf{x}_u(T-t-\delta)} \frac{\partial \mathbf{x}_u(T)}{\partial \mathbf{x}_u(T-t)} \\
&= \left(I_u + \int_0^{\delta} \frac{\partial \mathbf{y}(T-t-\delta+s)}{\partial \mathbf{x}_u(T-t-\delta)} \left.\frac{\partial f_u}{\partial \mathbf{y}}\right|_{\mathbf{y}(T-t-\delta+s)} \mathrm{d}s\right) \frac{\partial \mathbf{x}_u(T)}{\partial \mathbf{x}_u(T-t)}
\end{aligned} \quad (24)$$

This way, we have expressed $\frac{\partial \mathbf{x}_u(T)}{\partial \mathbf{x}_u(T-t-\delta)}$ in terms of $\frac{\partial \mathbf{x}_u(T)}{\partial \mathbf{x}_u(T-t)}$. To calculate our objective, we want to differentiate with respect to $\delta$. We first calculate the difference:

$$\frac{\partial \mathbf{x}_u(T)}{\partial \mathbf{x}_u(T-t-\delta)} - \frac{\partial \mathbf{x}_u(T)}{\partial \mathbf{x}_u(T-t)} = \left(\int_0^{\delta} \frac{\partial \mathbf{y}(T-t-\delta+s)}{\partial \mathbf{x}_u(T-t-\delta)} \left.\frac{\partial f_u}{\partial \mathbf{y}}\right|_{\mathbf{y}(T-t-\delta+s)} \mathrm{d}s\right) \frac{\partial \mathbf{x}_u(T)}{\partial \mathbf{x}_u(T-t)} \quad (25)$$

We can now divide by $\delta$ and take the limit $\delta \to 0$

$$\begin{aligned}
&\lim_{\delta \to 0} \frac{1}{\delta} \left(\frac{\partial \mathbf{x}_u(T)}{\partial \mathbf{x}_u(T-t-\delta)} - \frac{\partial \mathbf{x}_u(T)}{\partial \mathbf{x}_u(T-t)}\right) \\
&= \lim_{\delta \to 0} \left(\frac{1}{\delta} \int_0^{\delta} \frac{\partial \mathbf{y}(T-t-\delta+s)}{\partial \mathbf{x}_u(T-t-\delta)} \left.\frac{\partial f_u}{\partial \mathbf{y}}\right|_{\mathbf{y}(T-t-\delta+s)} \mathrm{d}s\right) \frac{\partial \mathbf{x}_u(T)}{\partial \mathbf{x}_u(T-t)} \\
&= \left.\frac{\partial \mathbf{y}}{\partial \mathbf{x}_u}\right|_{(T-t)} \left.\frac{\partial f_u}{\partial \mathbf{y}}\right|_{\mathbf{y}(T-t)} \frac{\partial \mathbf{x}_u(T)}{\partial \mathbf{x}_u(T-t)}
\end{aligned} \quad (26)$$

Where in the final equality we used the fundamental theorem of calculus. Finally

$$\frac{\mathrm{d}}{\mathrm{d}t} \frac{\partial \mathbf{x}_u(T)}{\partial \mathbf{x}_u(T-t)} = \left.\frac{\partial \mathbf{y}}{\partial \mathbf{x}_u}\right|_{(T-t)} \left.\frac{\partial f_u}{\partial \mathbf{y}}\right|_{\mathbf{y}(T-t)} \frac{\partial \mathbf{x}_u(T)}{\partial \mathbf{x}_u(T-t)} = F_u \frac{\partial \mathbf{x}_u(T)}{\partial \mathbf{x}_u(T-t)} \quad (27)$$

giving us the final result. □

We are now ready to prove Theorem 2.3. First, we calculate that

$$\frac{\partial f_u}{\partial \mathbf{y}} = \frac{\partial}{\partial y} \left(L_u \mathcal{J} \frac{\partial H}{\partial \mathbf{y}}\right) = \frac{\partial^2 H}{\partial \mathbf{y}^2} \mathcal{J}^{\top} L_u^{\top} = S \mathcal{J}^{\top} L_u^{\top} \quad (28)$$

so that $F_u = L_u S \mathcal{J}^{\top} L_u^{\top}$. This will be helpful in the following matrix calculations.

*Proof.* For brevity, we call $\left[\frac{\partial \mathbf{x}_u(T)}{\partial \mathbf{x}_u(T-t)}\right] = \Psi_u(T, T-t)$, which will be indicated simply as $\Psi_u$. When $t = 0$, $\Psi_u$ is just the Jacobian of the identity map $\Psi_u(T, T) = I_u$ and the result $\Psi_u^\top \mathcal{J}_u \Psi_u = \mathcal{J}_u$ is true for $t = 0$. Calculating the time derivative on $\Psi_u^\top \mathcal{J}_u \Psi_u$ we have that

$$
\begin{aligned}
\frac{\mathrm{d}}{\mathrm{d}t}[\Psi_u^\top \mathcal{J}_u \Psi_u] &= \dot{\Psi}_u^\top \mathcal{J}_u \Psi_u + \Psi_u^\top \mathcal{J}_u \dot{\Psi}_u \\
&= (F_u \Psi_u)^\top \mathcal{J}_u \Psi_u + \Psi_u^\top \mathcal{J}_u F_u \Psi_u = \\
&= \Psi_u^\top L_u \mathcal{J} S^\top L_u^\top \mathcal{J}_u \Psi_u + \Psi_u^\top \mathcal{J}_u L_u S \mathcal{J}^\top L_u^\top \Psi \\
&= \Psi_u^\top \left( L_u \mathcal{J} S L_u^\top \mathcal{J}_u + \mathcal{J}_u L_u S \mathcal{J}^\top L_u^\top \right) \Psi_u
\end{aligned}
\tag{29}
$$

where in the second equality we used the result from lemma B.1. We just need to show that the term in parentheses is zero so that the time derivative is zero. Using the relations $\mathcal{J}^\top L_u^\top = -L_u^\top \mathcal{J}_u$ and $J_u L_u = L_u \mathcal{J}$ we easily see that, finally

$$
\frac{\mathrm{d}}{\mathrm{d}t}\left( \left[\frac{\partial \mathbf{x}_u(T)}{\partial \mathbf{x}_u(T-t)}\right]^\top \mathcal{J}_u \left[\frac{\partial \mathbf{x}_u(T)}{\partial \mathbf{x}_u(T-t)}\right] \right) = \Psi_u^\top \left( L_u \mathcal{J} S L_u^\top \mathcal{J}_u + L_u \mathcal{J} S(-L_u^\top \mathcal{J}_u) \right) \Psi_u = 0
\tag{30}
$$

which means that $\left[\frac{\partial \mathbf{x}_u(T)}{\partial \mathbf{x}_u(T-t)}\right]^\top \mathcal{J}_u \left[\frac{\partial \mathbf{x}_u(T)}{\partial \mathbf{x}_u(T-t)}\right]$ is constant and equal to $\mathcal{J}_u$ for all $t$, that is our thesis. Now, the bound on the gradient follows by considering any sub-multiplicative norm $\|\cdot\|$:

$$
\|\mathcal{J}_u\| = \left\| \left[\frac{\partial \mathbf{x}_u(T)}{\partial \mathbf{x}_u(T-t)}\right]^\top \mathcal{J}_k \left[\frac{\partial \mathbf{x}_u(T)}{\partial \mathbf{x}_u(T-t)}\right] \right\| \leq \left\| \frac{\partial \mathbf{x}_u(T)}{\partial \mathbf{x}_u(T-t)} \right\|^2 \|\mathcal{J}_u\|
$$

and simplifying by $\|\mathcal{J}_u\| = 1$. $\qquad\square$

### B.4 PROOF OF THEOREM A.1

To prove the upper bound, we use the following technical lemma:

**Lemma B.2** (Galimberti et al. (2023)). *Consider a matrix $\mathbf{A} \in \mathbb{R}^{n \times n}$ with columns $\mathbf{a}_i \in \mathbb{R}^n$, i.e., $\mathbf{A} = \begin{bmatrix} \mathbf{a}_1 & \mathbf{a}_2 & \cdots & \mathbf{a}_n \end{bmatrix}$, and assume that $\|\mathbf{a}_i\|_2 \leq \gamma^+$ for all $i = 1, \ldots, n$. Then, $\|\mathbf{A}\|_2 \leq \gamma^+ \sqrt{n}$.*

This lemma gives a bound on the spectral norm of a matrix when its columns are uniformly bounded in norm. Therefore, our proof strategy for Theorem A.1 lies in bounding each column of the BSM matrix.

*Proof.* Consider the ODE in Eq. (27) from Lemma B.1 and split $\frac{\partial \mathbf{x}_u(T)}{\partial \mathbf{x}_u(T-t)}$ into columns $\frac{\partial \mathbf{x}_u(T)}{\partial \mathbf{x}_u(T-t)} = \begin{bmatrix} \mathbf{z}_1(\mathbf{t}) & \mathbf{z}_2(t) & \ldots & \mathbf{z}_d(t) \end{bmatrix}$. Then, Eq. (27) is equivalent to

$$
\dot{\mathbf{z}}_i(t) = \mathbf{A}_u(T-t)\mathbf{z}_i(t), \quad t \in [0, T], i = 1, 2 \ldots, d,
\tag{31}
$$

subject to $\mathbf{z}_i(0) = e_i$, where $e_i$ is the unit vector with a single nonzero entry in position $i$. The solution of the linear system of ODEs in Eq. (31) is given by the integral equation

$$
\mathbf{z}_i(t) = \mathbf{z}_i(0) + \int_0^t \mathbf{A}_u(T-s)\mathbf{z}_i(s)ds, \quad t \in [0, T].
\tag{32}
$$

By assuming that $\|\mathbf{A}_u(\tau)\|_2 \leq Q$ for all $\tau \in [0, T]$, and applying the triangular inequality in Eq. (32), it is obtained that:

$$
\|\mathbf{z}_i(t)\|_2 \leq \|\mathbf{z}_i(0)\|_2 + Q \int_0^t \|\mathbf{z}_i(s)\|_2 \, ds = 1 + Q \int_0^t \|\mathbf{z}_i(s)\|_2 \, ds,
$$

where the last equality follows from $\|\mathbf{z}_i(0)\|_2 = \|e_i\|_2 = 1$ for all $i = 1, 2, \ldots, d$. Then, applying the Gronwall inequality, it holds for all $t \in [0, T]$

$$
\|\mathbf{z}_i(t)\|_2 \leq \exp(QT).
\tag{33}
$$

By applying Lemma B.2, the general bound follows.

Lastly, we characterize $Q$ by bounding the norm $\|\mathbf{A}_u(\tau)\|_2 \; \forall \tau \in [0, T]$. From Lemma B.1 $\mathbf{A}_v$ can be expressed as $\mathbf{A}_u = \mathbf{L}_u \mathbf{S} \mathcal{J}^\top \mathbf{L}_u^\top$, which is equivalently $\mathbf{A}_u = \nabla^2_{\mathbf{x}_u} H_\mathcal{G}(\mathbf{y}) \mathcal{J}_u^\top$, since $\mathcal{J}^\top \mathbf{L}_u^\top = \mathbf{L}_v^\top \mathcal{J}_u^\top$. The Hessian $\nabla^2_{\mathbf{x}_u} H_\mathcal{G}(\mathbf{y})$ is of the form:

$$\nabla^2_{\mathbf{x}_u} H_\mathcal{G}(\mathbf{y}) = \mathbf{W}^\top \operatorname{diag}(\sigma'(\mathbf{W}\mathbf{x}_u + \Phi_u + b))\mathbf{W} + \sum_{v \in \mathcal{N}_u} \mathbf{V}^\top \operatorname{diag}(\sigma'(\mathbf{W}\mathbf{x}_v + \Phi_v + b))\mathbf{V}.$$

After noting that $\|\nabla^2_{\mathbf{x}_u} H_\mathcal{G}(\mathbf{y})\mathcal{J}_u^\top\|_2 \le \|\nabla^2_{\mathbf{x}_u} H_\mathcal{G}(\mathbf{y})\|_2\|\mathcal{J}_u^\top\|_2$, the only varying part is the Hessian $\nabla^2_{\mathbf{x}_u} H_\mathcal{G}(\mathbf{y})$ since $\|\mathcal{J}_u^\top\|_2 = 1$. By Lemma B.2 $\operatorname{diag}(\sigma'(x)) \le \sqrt{d}\, M$ and noting that $\|\mathbf{X}^\top\| = \|\mathbf{X}\|$ for any square matrix $\mathbf{X}$, then

$$\left\|\nabla^2_{\mathbf{x}_u} H_\mathcal{G}(\mathbf{y})\right\|_2 \le \sqrt{d}\, M\, \|\mathbf{W}\|_2^2 + \sqrt{d}\, M\, \max_{i \in [n]} |\mathcal{N}_i|\, \|\mathbf{V}\|_2^2 =: Q.$$

This also justifies our previous assumption that $\|\mathbf{A}_u(\tau)\|_2$ is bounded. $\qquad\square$

## B.5 PROOF OF THEOREM A.2

*Proof.* In the discrete case, since the semi-implicit Euler integration scheme is a symplectic method, it holds that:

$$\left[\frac{\partial \mathbf{x}_u^{(\ell)}}{\partial \mathbf{x}_u^{(\ell-1)}}\right]^\top \mathcal{J}_u \left[\frac{\partial \mathbf{x}_u^{(\ell)}}{\partial \mathbf{x}_u^{(\ell-1)}}\right] = \mathcal{J}_u \tag{34}$$

Further, by using the chain rule and applying Eq. (34) iteratively we get:

$$\left[\frac{\partial \mathbf{x}_u^{(L)}}{\partial \mathbf{x}_u^{(L-\ell)}}\right]^\top \mathcal{J}_u \left[\frac{\partial \mathbf{x}_u^{(L)}}{\partial \mathbf{x}_u^{(L-\ell)}}\right] = \left[\prod_{i=L-\ell}^{L-1} \frac{\partial \mathbf{x}_u^{(i+1)}}{\partial \mathbf{x}_u^{(i)}}\right]^\top \mathcal{J}_u \left[\prod_{i=L-\ell}^{L-1} \frac{\partial \mathbf{x}_u^{(i+1)}}{\partial \mathbf{x}_u^{(i)}}\right] = \mathcal{J}_u$$

Hence, the BSM is symplectic at arbitrary depth and we can conclude the proof with:

$$\|\mathcal{J}_u\| = \left\|\left[\frac{\partial \mathbf{x}_u^{(L)}}{\partial \mathbf{x}_u^{(L-\ell)}}\right]^\top \mathcal{J}_u \left[\frac{\partial \mathbf{x}_u^{(L)}}{\partial \mathbf{x}_u^{(L-\ell)}}\right]\right\| \le \left\|\frac{\partial \mathbf{x}_u^{(L)}}{\partial \mathbf{x}_u^{(L-\ell)}}\right\|^2 \|\mathcal{J}_u\|.$$

$$\square$$

## B.6 PROOF OF THEOREM 2.4

*Proof.* We employ a similar proof as in Di Giovanni et al. (2023) and redo some calculations based on our model and the Symplectic Euler scheme used to obtain Eqs. (12) and (13). To calculate $\frac{\partial \mathbf{x}_u^{(\ell+1)}}{\partial \mathbf{x}_v^{(\ell)}}$ for two consecutive nodes $u, v$, we consider the 4 sub-blocks as per

$$\frac{\partial \mathbf{x}_u^{(\ell+1)}}{\partial \mathbf{x}_v^{(\ell)}} = \begin{bmatrix} \frac{\partial \mathbf{p}_v^{(\ell+1)}}{\partial \mathbf{p}_v^{(\ell)}} & \frac{\partial \mathbf{q}_v^{(\ell+1)}}{\partial \mathbf{p}_v^{(\ell)}} \\ \frac{\partial \mathbf{p}_v^{(\ell+1)}}{\partial \mathbf{q}_v^{(\ell)}} & \frac{\partial \mathbf{q}_v^{(\ell+1)}}{\partial \mathbf{q}_v^{(\ell)}} \end{bmatrix}. \tag{35}$$

We proceed to calculate the 4 blocks independently, stopping at first-order terms in $\epsilon$ for simplicity reasons:

$$\begin{aligned}
\frac{\partial \mathbf{p}_u^{\ell+1}}{\partial \mathbf{p}_v^\ell} &= \sum_{w \in \mathcal{N}_u \cap \mathcal{N}_v} \frac{\partial \mathbf{q}_w^{\ell+1}}{\partial \mathbf{p}_v^\ell} \frac{\partial \mathbf{p}_u^{\ell+1}}{\partial \mathbf{q}_w^\ell} = O(\epsilon^2), \\
\frac{\partial \mathbf{p}_u^{\ell+1}}{\partial \mathbf{q}_v^\ell} &= -\epsilon \mathbf{A}_{uv}\left(\mathbf{V}_q^\top \mathbf{W}_q^\top \sigma_q'(u) + \mathbf{W}_q^\top \mathbf{V}_q^\top \sigma_q'(v) + \sum_{w \in \mathcal{N}_u \cap \mathcal{N}_v}(\mathbf{V}_q^\top)^2 \sigma_q'(w)\right), \\
\frac{\partial \mathbf{q}_u^{\ell+1}}{\partial \mathbf{q}_v^\ell} &= \sum_{w \in \mathcal{N}_u \cap \mathcal{N}_v} \frac{\partial \mathbf{p}_w^{\ell+1}}{\partial \mathbf{q}_v^\ell} \frac{\partial \mathbf{q}_u^{\ell+1}}{\partial \mathbf{p}_w^{\ell+1}} = O(\epsilon^2), \\
\frac{\partial \mathbf{q}_u^{\ell+1}}{\partial \mathbf{p}_v^\ell} &= \epsilon \mathbf{A}_{uv}\left(\mathbf{V}_p^\top \mathbf{W}_p^\top \sigma_p'(u) + \mathbf{W}_p^\top \mathbf{V}_p^\top \sigma_p'(v) + \sum_{w \in \mathcal{N}_u \cap \mathcal{N}_v}(\mathbf{V}_p^\top)^2 \sigma_p'(w)\right),
\end{aligned} \tag{36}$$

where compressed $\sigma_p(u) = \sigma(\mathbf{W}_p \mathbf{p}_u^{(\ell+1)} + \Phi_{\mathcal{G}}(\{\mathbf{p}_v^{(\ell+1)}\}_{v \in \mathcal{N}_u}))$ and analogously for $q$ for length reasons. Now, the norm of a block matrix is less or equal to the sum of the norms of the sub-blocks, so that

$$
\begin{aligned}
\left\| \frac{\partial \mathbf{x}_u^{(\ell+1)}}{\partial \mathbf{x}_v^{(\ell)}} \right\|_{L_1} &\leq \left\| \frac{\partial \mathbf{p}_u^{(\ell+1)}}{\partial \mathbf{p}_v^{(\ell)}} \right\|_{L_1} + \left\| \frac{\partial \mathbf{p}_u^{(\ell+1)}}{\partial \mathbf{q}_v^{(\ell)}} \right\|_{L_1} + \left\| \frac{\partial \mathbf{q}_u^{(\ell+1)}}{\partial \mathbf{p}_v^{(\ell)}} \right\|_{L_1} + \left\| \frac{\partial \mathbf{q}_u^{(\ell+1)}}{\partial \mathbf{q}_v^{(\ell)}} \right\|_{L_1} \\
&\leq \left\| \frac{\partial \mathbf{p}_u^{(\ell+1)}}{\partial \mathbf{q}_v^{(\ell)}} \right\|_{L_1} + \left\| \frac{\partial \mathbf{q}_u^{(\ell+1)}}{\partial \mathbf{p}_v^{(\ell)}} \right\|_{L_1} + O(\epsilon^2).
\end{aligned}
\tag{37}
$$

Now, using the Eq. (36) and setting $c_\sigma = \max_x |\sigma'(x)|$, we can estimate

$$
\begin{aligned}
\left\| \frac{\partial \mathbf{x}_u^{(\ell+1)}}{\partial \mathbf{x}_v^{(\ell)}} \right\|_{L_1} &\leq \epsilon \mathbf{A}_{uv} \left( 2 d c_\sigma |\mathbf{W}_q||\mathbf{V}_q| + \sum_w \mathbf{A}_{vw} \mathbf{A}_{wu} |\mathbf{V}_q^2| d c_\sigma \right) \\
&\quad + \epsilon \mathbf{A}_{uv} \left( 2 d c_\sigma |\mathbf{W}_p||\mathbf{V}_p| + \sum_w \mathbf{A}_{vw} \mathbf{A}_{wu} |\mathbf{V}_p^2| d c_\sigma \right).
\end{aligned}
\tag{38}
$$

Recalling the block structure of $\mathbf{V}$, we can say that $|\mathbf{V}| = |\mathbf{V}_p| + |\mathbf{V}_q|$ and obtain this final form of the one-step neighbor sensitivity:

$$
\left\| \frac{\partial \mathbf{x}_u^{(\ell+1)}}{\partial \mathbf{x}_v^{(\ell)}} \right\|_{L_1} \leq \epsilon \mathbf{A}_{uv} c_\sigma d \left( 2|\mathbf{V}||\mathbf{W}| + \sum_w \mathbf{A}_{vw} \mathbf{A}_{wu} |\mathbf{V}^2| \right).
\tag{39}
$$

Repeating the same process for the same node update, one has that

$$
\left\| \frac{\partial \mathbf{x}_v^{(\ell+1)}}{\partial \mathbf{x}_v^{(\ell)}} \right\|_{L_1} = d \mathbf{I} \left( 1 + \epsilon c_\sigma \left( |\mathbf{W}^2| + \sum_w \mathbf{A}_{uw} \mathbf{A}_{wu} |\mathbf{V}^2| \right) \right).
\tag{40}
$$

Now, we can show the inductive step:

$$
\frac{\partial \mathbf{x}_u^{(\ell+1)}}{\partial \mathbf{x}_v^{(0)}} = \frac{\partial \mathbf{x}_u^{(\ell)}}{\partial \mathbf{x}_v^{(0)}} \frac{\partial \mathbf{x}_u^{(\ell+1)}}{\partial \mathbf{x}_u^{(\ell)}} + \sum_{w \in \mathcal{N}_u} \frac{\partial \mathbf{x}_w^{(\ell)}}{\partial \mathbf{x}_v^{(0)}} \frac{\partial \mathbf{x}_u^{(\ell+1)}}{\partial \mathbf{x}_w^{(\ell)}}.
\tag{41}
$$

We now evaluate the norm of this object and, by using the triangular inequality and the two relations above for same-node and neighboring-node updates, we obtain

$$
\begin{aligned}
\left\| \frac{\partial \mathbf{x}_u^{(\ell+1)}}{\partial \mathbf{x}_v^{(0)}} \right\|_{L_1} &\leq \left\| \frac{\partial \mathbf{x}_u^{(\ell)}}{\partial \mathbf{x}_v^{(0)}} \right\|_{L_1} \left\| \frac{\partial \mathbf{x}_u^{(\ell+1)}}{\partial \mathbf{x}_u^{(\ell)}} \right\|_{L_1} + \sum_{w \in \mathcal{N}_u} \left\| \frac{\partial \mathbf{x}_w^{(\ell)}}{\partial \mathbf{x}_v^{(0)}} \right\|_{L_1} \left\| \frac{\partial \mathbf{x}_u^{(\ell+1)}}{\partial \mathbf{x}_w^{(\ell)}} \right\|_{L_1} \\
&\leq \left\| \frac{\partial \mathbf{x}_u^{(\ell)}}{\partial \mathbf{x}_v^{(0)}} \right\|_{L_1} d \mathbf{I} \left( 1 + \epsilon c_\sigma |\mathbf{W}^2| + \epsilon c_\sigma \sum_w \mathbf{A}_{uw} \mathbf{A}_{wu} |\mathbf{V}^2| \right) \\
&\quad + d \mathbf{A}_{uv} \epsilon c_\sigma \sum_w \mathbf{A}_{uw} \left\| \frac{\partial \mathbf{x}_w^{(\ell)}}{\partial \mathbf{x}_v^{(0)}} \right\|_{L_1} \left( 2|\mathbf{V}||\mathbf{W}| + \sum_z \mathbf{A}_{vz} \mathbf{A}_{zw} |\mathbf{V}^2| \right).
\end{aligned}
\tag{42}
$$

If we now consider the upper bound to be independent of the specific nodes and the maximum degree of a node to be $N$, we have that

$$
\begin{aligned}
\left\| \frac{\partial \mathbf{x}_u^{(\ell+1)}}{\partial \mathbf{x}_v^{(0)}} \right\|_{L_1} &\leq \left\| \frac{\partial \mathbf{x}_u^{(\ell)}}{\partial \mathbf{x}_v^{(0)}} \right\|_{L_1} d \mathbf{I} \left( 1 + \epsilon c_\sigma |\mathbf{W}^2| + \epsilon c_\sigma \sum_w \mathbf{A}_{uw} \mathbf{A}_{wu} |\mathbf{V}^2| \right) \\
&\quad + \left\| \frac{\partial \mathbf{x}_u^{(\ell)}}{\partial \mathbf{x}_v^{(0)}} \right\|_{L_1} \mathbf{A}_{uv} N d \epsilon c_\sigma \left( 2|\mathbf{V}||\mathbf{W}| + \sum_z \mathbf{A}_{vz} \mathbf{A}_{zu} |\mathbf{V}^2| \right) \\
&\leq \left\| \frac{\partial \mathbf{x}_u^{(\ell)}}{\partial \mathbf{x}_v^{(0)}} \right\|_{L_1} \mathbf{I} d \left( 1 + \epsilon c_\sigma |\mathbf{W}^2| + (N-1)\epsilon|\mathbf{V}^2| \right) \\
&\quad + \left\| \frac{\partial \mathbf{x}_u^{(\ell)}}{\partial \mathbf{x}_v^{(0)}} \right\|_{L_1} \mathbf{A}_{uv} d \left( N \epsilon c_\sigma \left( 2|\mathbf{V}||\mathbf{W}| + (N-1)|\mathbf{V}^2| \right) \right).
\end{aligned}
\tag{43}
$$

which is in the form of theorem 3.2 of Di Giovanni et al. (2023). Calling $w = \max\{|W|, |V|\}$, we obtain that

$$\left\|\frac{\partial \mathbf{x}_u^{(\ell+1)}}{\partial \mathbf{x}_v^{(0)}}\right\|_{L_1} \leq \left\|\frac{\partial \mathbf{x}_u^{(\ell)}}{\partial \mathbf{x}_v^{(0)}}\right\|_{L_1} \left(\mathbf{I}d(1 + \epsilon c_\sigma N w^2) + \mathbf{A}_{uv} dw^2 \epsilon c_\sigma (N^2 + N)\right). \tag{44}$$

We now sacrifice the term 1 and set $\epsilon = 1$ we can collect a term $w c_\sigma N d$ and, following proof in Di Giovanni et al. (2023), we finally arrive at

$$\left\|\frac{\partial \mathbf{x}_u^{(\ell+1)}}{\partial \mathbf{x}_v^{(0)}}\right\|_{L_1} \leq (dwNc_\sigma)^{\ell+1}((w\mathbf{I} + w(N+1)\mathbf{A})^{\ell+1})_{uv}. \tag{45}$$

Without the terms we sacrificed along the proof, our upper bound is at least $N^\ell$ greater than the one in Di Giovanni et al. (2023). □

### B.7 ADDITIONAL THEOREMS FOR PH-DGN WITH DRIVING FORCES

In this section, we consider the case of $\sigma$ being a bounded activation function $|\sigma(\mathbf{x})| \leq b_\sigma$ with a bounded derivative $|\sigma'(\mathbf{x})| \leq c_\sigma$, e.g. $\tanh$. Further, we suppose the driving forces are bounded and Lipschitz continuous with common constant $B_d$. In formulas, this means that $|D_u(\mathbf{q})| \leq B_d$, $\left|\frac{\partial D_u(\mathbf{q})}{\partial \mathbf{q}}\right| \leq B_d$, $|F(\mathbf{q}, t)| \leq B_d$ and $\left|\frac{\partial F(\mathbf{q},t)}{\partial \mathbf{q}}\right| \leq B_d$. Under these hypotheses, the self-influence of a node after one step is bounded from below by a constant. An upper bound on the backward sensitivity matrix (BSM) norm can also be derived for the same node influence and neighboring node influence after one step.

**Theorem B.1** (Lower bound, port-Hamiltonian case). *Consider the full port-Hamiltonian update in Eqs.* (13) *and* (14). *Then, with the above hypotheses, the following lower bound for the BSM holds*

$$\left\|\frac{\partial \mathbf{x}_u^{(\ell+1)}}{\partial \mathbf{x}_u^{(\ell)}}\right\|_{L_1} \geq \left|\left\|\frac{\partial \tilde{\mathbf{x}}_u^{(\ell+1)}}{\partial \mathbf{x}_u^{(\ell)}}\right\| - \|\mathbf{D}\|\right| \tag{46}$$
$$\geq 1 - \epsilon w B_d c_\sigma'(N + 3 + w),$$

*where $w = \max\{|\mathbf{W}|, |\mathbf{V}|\}$ as before. In this equation, $[\partial \mathbf{x}_u^{(\ell+1)}/\partial \mathbf{x}_v^{(\ell)}]$ represents the BSM for the full port-Hamiltonian update as in Eqs.* (13) *and* (14)*, while $[\partial \tilde{\mathbf{x}}_u^{(\ell+1)}/\partial \mathbf{x}_v^{(\ell)}]$ is the BSM for the pure Hamiltonian case as in Eqs.* (12) *and* (13)*, and $\mathbf{D}$ is defined as*

$$\mathbf{D} = \frac{\partial}{\partial \mathbf{x}_u^\ell}\left(D_u(\mathbf{q}^{(\ell+1)})\nabla_{\mathbf{p}_u} H_{\mathcal{G}}(\mathbf{p}^{(\ell)}, \mathbf{q}^{(\ell)})\right) + \frac{\partial F_u(\mathbf{q}^{(\ell)}, t)}{\partial \mathbf{x}_u^\ell} \tag{47}$$

This theorem shows that, with these hypotheses, the effect of the driving forces on the BSM norm can be controlled with a small enough step size.

*Proof.* We start by providing the same calculations as Theorem 2.4, this time with port-Hamiltonian components included, for the one-step update in both the cross-node case

$$\frac{\partial \mathbf{p}_u^{\ell+1}}{\partial \mathbf{p}_v^\ell} = -\epsilon \mathbf{A}_{\mathbf{uv}} D_u(\mathbf{q}_u)\left(\mathbf{V}_p^\top \mathbf{W}_p^\top \sigma_p'(u) + \mathbf{W}_p^\top \mathbf{V}_p^\top \sigma'(p)_v + \sum_{w \in \mathcal{N}_u \cap \mathcal{N}_v}(\mathbf{V}_p^\top)^2 \sigma_p'(w)\right) + O(\epsilon^2),$$

$$\frac{\partial \mathbf{p}_u^{\ell+1}}{\partial \mathbf{q}_v^\ell} = -\epsilon \mathbf{A}_{uv}\left(\mathbf{V}_q^\top \mathbf{W}_q^\top \sigma_q'(u) + \mathbf{W}_q^\top \mathbf{V}_q^\top \sigma_q'(v) + \sum_{w \in \mathcal{N}_u \cap \mathcal{N}_v}(\mathbf{V}_q^\top)^2 \sigma_q'(w)\right)$$
$$- \epsilon \mathbf{A}_{uv}\left(\frac{\partial D_u(\mathbf{q})}{\partial \mathbf{q}_v}(\mathbf{W}_p^\top \sigma_p(u) + \sum_{w \in \mathcal{N}_u} \mathbf{V}_p^\top \sigma_p(w)) + \frac{\partial F(\mathbf{q}, t)}{\partial \mathbf{q}_v}\right)$$

$$\frac{\partial \mathbf{q}_u^{\ell+1}}{\partial \mathbf{q}_v^\ell} + O(\epsilon^2) = \sum_{w \in \mathcal{N}_u \cap \mathcal{N}_v} \frac{\partial \mathbf{p}_w^{\ell+1}}{\partial \mathbf{q}_v^\ell}\frac{\partial \mathbf{q}_u^{\ell+1}}{\partial \mathbf{p}_w^{\ell+1}} = O(\epsilon^2),$$

$$\frac{\partial \mathbf{q}_u^{\ell+1}}{\partial \mathbf{p}_v^\ell} = \epsilon \mathbf{A}_{uv}\left(\mathbf{V}_p^\top \mathbf{W}_p^\top \sigma_p'(u) + \mathbf{W}_p^\top \mathbf{V}_p^\top \sigma_p'(v) + \sum_{w \in \mathcal{N}_u \cap \mathcal{N}_v}(\mathbf{V}_p^\top)^2 \sigma_p'(w)\right) + O(\epsilon^2) \tag{48}$$

and same node case

$$\frac{\partial \mathbf{p}_u^{\ell+1}}{\partial \mathbf{p}_u^\ell} = \mathbf{I} - \epsilon \left( D_u(\mathbf{q})(\mathbf{W}_p^\top)^2 \sigma_p(u) \right) + O(\epsilon^2),$$

$$\frac{\partial \mathbf{p}_u^{\ell+1}}{\partial \mathbf{q}_u^\ell} = -\epsilon \mathbf{I} \left( (\mathbf{W}_q^\top)^2 \sigma_q'(u) + \sum_{w \in \mathcal{N}_u} (\mathbf{V}_q^\top)^2 \sigma_q'(w) \right),$$

$$- \epsilon \mathbf{I} \left( \frac{\partial D_u(\mathbf{q})}{\partial \mathbf{q}_u} (\mathbf{W}_p^\top \sigma_p(u) + \sum_{w \in \mathcal{N}_u} \mathbf{V}_p^\top \sigma_p(w)) + \frac{\partial F(\mathbf{q}, t)}{\partial \mathbf{q}_u} \right) + O(\epsilon^2) \quad (49)$$

$$\frac{\partial \mathbf{q}_u^{\ell+1}}{\partial \mathbf{q}_u^\ell} = -\epsilon \mathbf{I} \left( (\mathbf{W}_p^\top)^2 \sigma_p'(u) + \sum_{w \in \mathcal{N}_u} (\mathbf{V}_p^\top)^2 \sigma_p'(w) \right) + O(\epsilon^2),$$

$$\frac{\partial \mathbf{q}_u^{\ell+1}}{\partial \mathbf{p}_u^\ell} = \mathbf{I} + O(\epsilon^2).$$

If we consider the one-step update for the same node, we can divide it into the non-dissipative component, called $\frac{\partial \tilde{\mathbf{x}}_u^{(\ell+1)}}{\partial \mathbf{x}_u^{(\ell)}}$, and the dissipative component given by

$$\mathbf{D} = -\epsilon \left( D_u(\mathbf{q})(\mathbf{W}_p^\top)^2 \sigma_p(u) \right) - \epsilon \mathbf{I} \left( \frac{\partial D_u(\mathbf{q})}{\partial \mathbf{q}_u} (\mathbf{W}_p^\top \sigma_p(u) + \sum_{w \in \mathcal{N}_u} \mathbf{V}_p^\top \sigma_p(w)) + \frac{\partial F(\mathbf{q}, t)}{\partial \mathbf{q}_u} \right)$$
$$(50)$$

Now, to estimate a lower bound on the self-influence norm, we can use the triangular inequality

$$\left\| \frac{\partial \mathbf{x}_u^{(\ell+1)}}{\partial \mathbf{x}_u^{(\ell)}} \right\|_{L_1} \geq \left| \left\| \frac{\partial \tilde{\mathbf{x}}_u^{(\ell+1)}}{\partial \mathbf{x}_u^{(\ell)}} \right\|_{L_1} - \|\mathbf{D}\|_{L_1} \right| \quad (51)$$

Since the first term is the update for the non-dissipative case, Theorem A.2 holds and then the right-hand side is greater or equal to

$$(1 - \|\mathbf{D}\|). \quad (52)$$

The driving forces effect can then be estimated as

$$\|\mathbf{D}\| \leq \epsilon(\|\mathbf{W}\|^2 |\sigma'| |D_u|) + \epsilon L(1 + \|\mathbf{W}\| |\sigma| + N \|\mathbf{V}\| |\sigma|)$$
$$\leq d\epsilon w L c_\sigma'(N + 3 + w) \quad (53)$$

as per the definitions above, which leads to Eq. (46). □

We now calculate an upper bound on the one-step BSM on a node $u$.

**Theorem B.2** (BSM upper bound, same-node update in port-Hamiltonian case). *Consider again the full port-Hamiltonian update in Eqs. (13) and (14) with the same hypothesis above. Then, the following upper bound for the BSM holds:*

$$\left\| \frac{\partial \mathbf{x}_u^{(\ell+1)}}{\partial \mathbf{x}_u^{(\ell)}} \right\|_{L_1} \leq \left\| \frac{\partial \tilde{\mathbf{x}}_u^{(\ell+1)}}{\partial \mathbf{x}_u^{(\ell)}} \right\| + \|\mathbf{D}\|$$
$$\leq d(1 + \epsilon c_\sigma' w(1 + B_d)(N + w + 3))), \quad (54)$$

*where the individual terms are defined as in theorem B.1.*

*Proof.* We start by recalling the calculations for the same-node and cross-node update gradients from Theorem B.1. Then, we employ again the triangular inequality to obtain

$$\left\| \frac{\partial \mathbf{x}_u^{(\ell+1)}}{\partial \mathbf{x}_u^{(\ell)}} \right\|_{L_1} \leq \left\| \frac{\partial \tilde{\mathbf{x}}_u^{(\ell+1)}}{\partial \mathbf{x}_u^{(\ell)}} \right\|_{L_1} + \|\mathbf{D}\|_{L_1} \quad (55)$$

Since the first term involves the non-dissipative components only, we use the same calculations as in Theorem 2.4. We use the same calculations for the second term as in Theorem B.1. Finally, we can calculate

$$
\begin{aligned}
\left\| \frac{\partial \mathbf{x}_u^{(\ell+1)}}{\partial \mathbf{x}_u^{(\ell)}} \right\|_{L_1} &\leq \left\| \frac{\partial \tilde{\mathbf{x}}_u^{(\ell+1)}}{\partial \mathbf{x}_u^{(\ell)}} \right\|_{L_1} + \|\mathbf{D}\|_{L_1} \\
&\leq \mathbf{I}d\left(1 + \epsilon c_\sigma |\mathbf{W}^2| + (N-1)\epsilon|\mathbf{V}^2|\right) + d\epsilon w L c'_\sigma (N+3+w) \\
&\leq d(1 + \epsilon c'_\sigma N w) + d\epsilon w L c'_\sigma (N+3+w) \\
&\leq d(1 + \epsilon c'_\sigma w(1+L)(N+w+3)))
\end{aligned}
\tag{56}
$$

$\square$

**Theorem B.3** (BSM upper bound, cross-node update in the port-Hamiltonian case). *Consider again the full port-Hamiltonian update in Eqs. (13) and (14) with the same hypothesis above. Then, the following upper bound for the cross-node BSM holds:*

$$
\begin{aligned}
\left\| \frac{\partial \mathbf{x}_u^{(\ell+1)}}{\partial \mathbf{x}_v^{(\ell)}} \right\|_{L_1} &\leq \left\| \frac{\partial \tilde{\mathbf{x}}_u^{(\ell+1)}}{\partial \mathbf{x}_v^{(\ell)}} \right\| + \|\mathbf{D}_{cross}\| \\
&\leq d(\epsilon c'_\sigma w(1+L)(N+w+3))),
\end{aligned}
\tag{57}
$$

*where $[\partial \mathbf{x}_u^{(\ell+1)}/\partial \mathbf{x}_v^{(\ell)}]$ represents the cross-node BSM for the full port-Hamiltonian update as in Eqs. (13) and (14), while $[\partial \tilde{\mathbf{x}}_u^{(\ell+1)}/\partial \mathbf{x}_v^{(\ell)}]$ is the cross-node BSM for the pure Hamiltonian case as in Eqs. (12) and (13), and $\mathbf{D}_{cross}$ is defined as*

$$
\mathbf{D}_{cross} = \frac{\partial}{\partial \mathbf{x}_v^\ell}\left( D_u(\mathbf{q}^{(\ell+1)}) \nabla_{\mathbf{p}_u} H_\mathcal{G}(\mathbf{p}^{(\ell)}, \mathbf{q}^{(\ell)}) \right) + \frac{\partial F_u(\mathbf{q}^{(\ell)}, t)}{\partial \mathbf{x}_v^\ell}.
\tag{58}
$$

*Proof.* We start by recalling the calculations for the same-node and cross-node update gradients from Theorem B.1. Then, we employ again the triangular inequality to obtain

$$
\left\| \frac{\partial \mathbf{x}_u^{(\ell+1)}}{\partial \mathbf{x}_v^{(\ell)}} \right\|_{L_1} \leq \left\| \frac{\partial \tilde{\mathbf{x}}_u^{(\ell+1)}}{\partial \mathbf{x}_v^{(\ell)}} \right\|_{L_1} + \|\mathbf{D}_{cross}\|_{L_1}
\tag{59}
$$

Since the first term involves the non-dissipative components only, we use the same calculations as in Theorem 2.4. For the driving forces component, similar calculations to the ones in Theorem B.1 result in

$$
\|\mathbf{D}_{cross}\| \leq d(\epsilon c'_\sigma w(1+L)(N+w+3)))
\tag{60}
$$

from which our bound follows. $\square$

## C EXPERIMENTAL DETAILS

### C.1 EMPLOYED BASELINES

In our experiments, the performance of our port-Hamiltonian DGNs is compared with various state-of-the-art DGN baselines from the literature. Specifically, we consider:

- classical MPNN-based methods, i.e., GCN (Kipf & Welling, 2017), GraphSAGE (Hamilton et al., 2017), GAT (Veličković et al., 2018), GatedGCN (Bresson & Laurent, 2017), GIN (Xu et al., 2019), GINE (Hu et al., 2020), and GCNII (Chen et al., 2020);
- DE-DGNs, i.e., DGC (Wang et al., 2021), GRAND (Chamberlain et al., 2021a), Graph-CON (Rusch et al., 2022), A-DGN (Gravina et al., 2023), HANG (Zhao et al., 2023), HamGNN (Han et al., 2024) , and SWAN (Gravina et al., 2025a);
- Graph Transformers, i.e., Transformer (Vaswani et al., 2017; Dwivedi & Bresson, 2021), SAN (Kreuzer et al., 2021), and GraphGPS (Rampášek et al., 2022);
- Higher-Order DGNs, i.e., DIGL (Gasteiger et al., 2019), MixHop (Abu-El-Haija et al., 2019), and DRew (Gutteridge et al., 2023).

Note that since DE-DGNs belong to the same family as our proposed method, they are a direct competitor to our port-Hamiltonian DGNs.

## C.2 GRAPH TRANSFER TASK

The graph transfer task consists of the problem of propagating a label from a source node to a target node located at increased distance $k$ in the graph. In other words, we set up a node-level regression task that measures how much of the source node information has reached the target node. The task was first proposed in the work of (Di Giovanni et al., 2023), but here we follow the data generation and more challenging experimental setting of Gravina et al. (2025a). The task employ the same graph distributions as in Di Giovanni et al. (2023), i.e., line, ring, and crossed ring graphs (see Figure 4 for a visual exemplification of the three types of graphs when the distance between the source and target nodes is 5). However, to make the task more challenging we follow Gravina et al. (2025a), thus, we initialize node input features with random values uniformly sampled in the range $[0, 0.5)$. In each graph, the source node is assigned with label "1", and a target node is assigned with label "0". We use an input dimension of 1, and a source-target distance equal to $3, 5, 10$, and $50$. The target output is constructed by modifying the input features: the source node's target is the label "0", the target node's target is the label "1", and the intermediate nodes retain their original random values from the input. In other words, the ground truth values are the switched node labels of source and target nodes. We generate 1000 graphs for training, 100 for validation, and 100 for testing.

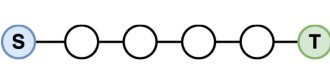 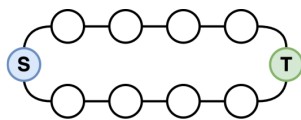 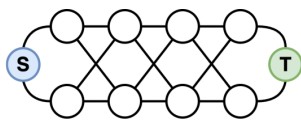

Figure 4: Three topologies for Graph Transfer. Left) Line. Center) Ring. Right) Crossed-Ring. The distance between source and target nodes is equal to 5. Nodes marked with $S$ are source nodes, while the nodes with a $T$ are target nodes.

Following Gravina et al. (2025a), We design each model using three main components. First is the encoder, which maps the node input features into a latent hidden space. Second is the graph convolution component (i.e., PH-DGN or other baselines). Third is the readout, which maps the convolution's output into the output space. The encoder and readout have the same architecture across all models in the experiments.

We perform hyperparameter tuning via grid search, optimizing the Mean Squared Error (MSE) computed on the node features of the whole graph. In other words, the model is trained to predict the target values for all nodes, and the loss is computed as the MSE between the predicted values and the corresponding target values across the entire graph. We train the models using Adam optimizer (Kingma & Ba, 2015) for a maximum of 2000 epochs and early stopping with patience of 100 epochs on the validation loss. For each model configuration, we perform 4 training runs with different weight initialization and report the average and standard deviation of the results. We report in Table 3 the grid of hyperparameter employed by each model in our experiment.

## C.3 GRAPH PROPERTY PREDICTION

The tasks consist of predicting two node-level (i.e., single source shortest path and eccentricity) and one graph-level (i.e., graph diameter) properties on synthetic graphs generated by different graph distributions. In our experiments, we follow the data generation and experimental procedure outlined in Gravina et al. (2023). Therefore, graphs contains between 25 and 35 nodes and are randomly generated from multiple distributions. Nodes are randomly initialized with a value uniformly sampled in the range $[0, 1)$, while target values represent single source shortest path, eccentricity, and graph diameter depending on the selected task. 5120 graphs are used as the training set, 640 as the validation set, and the rest as the test set.

As in the original work, each model is designed as three components, i.e., encoder, graph convolution component (PH-DGN or baselines), and readout. We perform hyperparameter tuning via grid search, optimizing the Mean Square Error (MSE), training the models using Adam optimizer for a maximum of 1500 epochs, and early stopping with patience of 100 epochs on the validation error. For each model configuration, we perform 4 training runs with different weight initialization and report the average of the results. We report in Appendix C.5 the grid of hyperparameters employed by each model in our experiment.

## C.4 LONG-RANGE GRAPH BENCHMARK

We consider the Peptide-func and Peptide-struct tasks from the Long-Range Graph Benchmark (Dwivedi et al., 2022). Both tasks contain graphs corresponding to peptide molecules for a total of 15,535 graphs, 2.3 million nodes and 4.7 million edges. Each graph has an average of 150 nodes, 307 edges, and an average diameter of 57. Peptide-func is a multi-label graph classification task with the aim of predicting the peptide function. Peptide-struct is a multi-dimensional graph regression task, whose objective is to predict structural properties of the peptides.

We follow the experimental setting of Dwivedi et al. (2022), thus we consider a stratified splitting of the data, with 70% of graphs as the training set, 15% as the validation set, and 15% as the test set. We perform hyperparameter tuning via grid search, optimizing the Average Precision (AP) in the Peptide-func task and the Mean Absolute Error (MAE) in the Peptide-struct task, training the models using AdamW optimizer for a maximum of 300 epochs. For each model configuration, we perform 3 training runs with different weight initialization and report the average of the results. Moreover, we stay within 500K parameter budget as proposed by Dwivedi et al. (2022). In our experiments, we also consider the setting of Tönshoff et al. (2023) that propose to employ a 3-layer MLP as readout. We report in Appendix C.5 the grid of hyperparameters employed by each model in our experiment.

## C.5 EXPLORED HYPERPARAMETER SPACE

In Table 3 we report the grid of hyperparameters employed in our experiments by each method, adhering to the established procedures for each task to ensure fair evaluation and reproducibility. We note that for hyperparameters specific to our PH-DGN, such as the step size $\epsilon$ and the number of layers $L$, we selected values based on the specific benchmark protocol (whenever possible) and considering factors like the average graph diameter in the training set.

Table 3: The grid of hyperparameters employed during model selection for the graph transfer tasks (*Transfer*), graph property prediction tasks (*GraphProp*), and Long Range Graph Benchmark (*LRGB*). We refer to Appendix A.2 for more details about dampening and external force implementations.

| Hyperparameters | Values | | |
|---|---|---|---|
| | *Transfer* | *GraphProp* | *LRGB* |
| Optimizer | Adam | Adam | AdamW |
| Learning rate | 0.001 | 0.003 | 0.001, 0.0005 |
| Weight decay | 0 | $10^{-6}$ | 0 |
| embedding dim | 64 | 10, 20, 30 | 195, 300 |
| N. layers ($L$) | 3, 5, 10, 50, 100, 150 | 1, 5, 10, 20, 30 | 32, 64 |
| Termination time ($T$) | $L\epsilon$ | 0.1, 1, 2, 3, 4 | 3, 5, 6 |
| $\epsilon$ | 0.5, 0.2, 0.1, 0.05, 0.01, $10^{-4}$ | $T/L$ | $T/L$ |
| $\Phi_{\mathbf{p}}$ | Eq. (6), GCN | Eq. (6), GCN | Eq. (6), GCN |
| $\Phi_{\mathbf{q}}$ | Eq. (6), GCN | Eq. (6), GCN | GCN |
| Readout input | $\mathbf{p}, \mathbf{q}, \mathbf{p}\|\mathbf{q}$ | $\mathbf{p}, \mathbf{q}, \mathbf{p}\|\mathbf{q}$ | $\mathbf{p}, \mathbf{q}, \mathbf{p}\|\mathbf{q}$ |
| $\sigma$ | tanh | tanh | tanh |
| Dampening | – | *param*, *param+*, *MLP4-ReLU*, *DGN-ReLU* | *param* |
| External Force | – | *MLP4-Sin*, *DGN-tanh* | *DGN-tanh* |
| N. readout layers | 1 | 1 | 1, 3 |

# D ADDITIONAL EXPERIMENTAL RESULTS

## D.1 GRAPH PROPERTY PREDICTION RUNTIMES

In Table 4 we report runtimes of PH-DGN with and without driving forces as well as baseline methods on the graph property prediction task in Section 3.3. To obtain a fair comparison, we configured all models with 20 layers and an embedding dimension of 30. We averaged the time (in seconds) over 4 random weight initializations for a complete epoch, which includes the forward and backward passes on the training data, as well as the forward pass on the validation and test data.

Table 4 shows that PH-DGN with and without driving forces has improved or comparable runtimes compared to MPNNs. Notably, PH-DGN without driving forces is on average 5.92 seconds faster than GAT and 5.19 seconds faster than GCN. Compared to DE-DGN baselines, our methods show longer execution times, which are inherently caused by the sequential computation of both $\mathbf{p}$ and $\mathbf{q}$ explicitly given in Appendix A.3 and non-conservative components (i.e., *param* and *MLP4-ReLU* in Table 4). Lastly, related Hamiltonian approaches require at least twice as much computation (HANG (Zhao et al., 2023)) up until 126x more seconds per epoch (HamGNN (Kang et al., 2023). This underlines our key contribution of a MPNN-complexity model adhering to Hamiltonian laws, i.e., linear in number of edges and nodes.

Table 4: Average time per epoch (measured in seconds) and std, averaged over 4 random weight initializations. Each time is obtained by employing 20 layers and an embedding dimension equal to 30. Our PH-DGN employs *param* and *MLP4-ReLU* as dampening and external force, respectively. The evaluation was carried out on an AMD EPYC 7543 CPU @ 2.80GHz. First, second, and **third** best results.

| Model | Diameter | SSSP | Eccentricity |
|---|---|---|---|
| **MPNNs** | | | |
| GCN | $32.45_{\pm 2.54}$ | $17.44_{\pm 3.85}$ | $11.78_{\pm 2.43}$ |
| GAT | $20.20_{\pm 5.18}$ | $26.41_{\pm 8.34}$ | $17.28_{\pm 1.92}$ |
| GraphSAGE | $13.12_{\pm 2.99}$ | $13.12_{\pm 2.99}$ | $\mathbf{8.20}_{\pm 0.75}$ |
| GIN | $6.63_{\pm 0.28}$ | $21.16_{\pm 2.33}$ | $14.22_{\pm 3.17}$ |
| GCNII | $13.13_{\pm 6.85}$ | $14.96_{\pm 7.17}$ | $15.70_{\pm 3.92}$ |
| **DE-DGNs** | | | |
| DGC | $\mathbf{8.97}_{\pm 9.07}$ | $12.54_{\pm 1.62}$ | $7.21_{\pm 11.10}$ |
| GRAND | $133.84_{\pm 42.57}$ | $109.15_{\pm 27.49}$ | $202.46_{\pm 85.01}$ |
| GraphCON | $9.26_{\pm 0.47}$ | $7.76_{\pm 0.05}$ | $7.80_{\pm 0.05}$ |
| A-DGN | $8.42_{\pm 2.71}$ | $7.86_{\pm 2.11}$ | $13.18_{\pm 9.07}$ |
| HamGNN | $1097.90_{\pm 379.56}$ | $1897.04_{\pm 22.08}$ | $1773.43_{\pm 54.37}$ |
| HANG | $28.92_{\pm 0.10}$ | $34.24_{\pm 1.08}$ | $34.63_{\pm 0.96}$ |
| **Ours** | | | |
| PH-DGN$_\mathrm{C}$ | $15.49_{\pm 0.05}$ | $15.28_{\pm 0.02}$ | $15.34_{\pm 0.04}$ |
| PH-DGN | $17.18_{\pm 0.04}$ | $17.12_{\pm 0.07}$ | $17.13_{\pm 0.06}$ |

## D.2 COMPLETE LRGB RESULTS

In Table 5 we report the complete results for the LRGB tasks including more multi-hop DGNs and ablating on the scores obtained with the original setting from Dwivedi et al. (2022) and the one proposed by Tönshoff et al. (2023), which leverage 3-layers MLP as a decoder to map the DGN output into the final prediction. Since we are unable to compare the validation scores of the baselines directly, we cannot determine the best method between (Dwivedi et al., 2022) and Tönshoff et al. (2023). Therefore, in Table 5, we present all the results.

Table 5: Results for Peptides-func and Peptides-struct averaged over 3 training seeds. Baseline results are taken from Dwivedi et al. (2022), Tönshoff et al. (2023), Gutteridge et al. (2023), Gravina et al. (2025a). "+PE/SE" indicates the use of positional or structural encoding. We have detailed the type of encoding wherever the original source explicitly specifies it. "*" means that we re-computed the performance of the method strictly following the experimental protocol from Tönshoff et al. (2023). As proposed in Tönshoff et al. (2023), re-evaluated methods employ a 3-layer MLP readout and (Laplacian) positional or (random walk) structural encoding. The **first**, **second**, and **third** best scores are colored.

| Model | Peptides-func AP ↑ | Peptides-struct MAE ↓ |
|---|---|---|
| **MPNNs, Dwivedi et al. (2022)** | | |
| GCN | $0.5930_{\pm 0.0023}$ | $0.3496_{\pm 0.0013}$ |
| GINE | $0.5498_{\pm 0.0079}$ | $0.3547_{\pm 0.0045}$ |
| GCNII | $0.5543_{\pm 0.0078}$ | $0.3471_{\pm 0.0010}$ |
| GatedGCN | $0.5864_{\pm 0.0077}$ | $0.3420_{\pm 0.0013}$ |
| **Multi-hop DGNs, Gutteridge et al. (2023)** | | |
| DIGL+MPNN | $0.6469_{\pm 0.0019}$ | $0.3173_{\pm 0.0007}$ |
| DIGL+MPNN+LapPE | $0.6830_{\pm 0.0026}$ | $0.2616_{\pm 0.0018}$ |
| MixHop-GCN | $0.6592_{\pm 0.0036}$ | $0.2921_{\pm 0.0023}$ |
| MixHop-GCN+LapPE | $0.6843_{\pm 0.0049}$ | $0.2614_{\pm 0.0023}$ |
| DRew-GCN | $0.6996_{\pm 0.0076}$ | $0.2781_{\pm 0.0028}$ |
| DRew-GCN+LapPE | $0.7150_{\pm 0.0044}$ | $0.2536_{\pm 0.0015}$ |
| DRew-GIN | $0.6940_{\pm 0.0074}$ | $0.2799_{\pm 0.0016}$ |
| DRew-GIN+LapPE | $0.7126_{\pm 0.0045}$ | $0.2606_{\pm 0.0014}$ |
| DRew-GatedGCN | $0.6733_{\pm 0.0094}$ | $0.2699_{\pm 0.0018}$ |
| DRew-GatedGCN+LapPE | $0.6977_{\pm 0.0026}$ | $0.2539_{\pm 0.0007}$ |
| **Transformers, Gutteridge et al. (2023)** | | |
| Transformer+LapPE | $0.6326_{\pm 0.0126}$ | $0.2529_{\pm 0.0016}$ |
| SAN+LapPE | $0.6384_{\pm 0.0121}$ | $0.2683_{\pm 0.0043}$ |
| GraphGPS+LapPE | $0.6535_{\pm 0.0041}$ | $0.2500_{\pm 0.0005}$ |
| **Modified experimental protocol, Tönshoff et al. (2023)** | | |
| GCN+PE/SE | $0.6860_{\pm 0.0050}$ | $0.2460_{\pm 0.0007}$ |
| GINE+PE/SE | $0.6621_{\pm 0.0067}$ | $0.2473_{\pm 0.0017}$ |
| GCNII+PE/SE* | $0.6444_{\pm 0.0011}$ | $0.2507_{\pm 0.0012}$ |
| GatedGCN+PE/SE | $0.6765_{\pm 0.0047}$ | $0.2477_{\pm 0.0009}$ |
| DRew-GCN+PE/SE* | $0.6945_{\pm 0.0021}$ | $0.2517_{\pm 0.0011}$ |
| GraphGPS+PE/SE | $0.6534_{\pm 0.0091}$ | $0.2509_{\pm 0.0014}$ |
| **DE-DGNs** | | |
| GRAND | $0.5789_{\pm 0.0062}$ | $0.3418_{\pm 0.0015}$ |
| GraphCON | $0.6022_{\pm 0.0068}$ | $0.2778_{\pm 0.0018}$ |
| A-DGN | $0.5975_{\pm 0.0044}$ | $0.2874_{\pm 0.0021}$ |
| SWAN | $0.6751_{\pm 0.0039}$ | $\mathbf{0.2485}_{\pm 0.0009}$ |
| **Ours** | | |
| PH-DGN$_C$ | $0.6961_{\pm 0.0070}$ | $0.2581_{\pm 0.0020}$ |
| PH-DGN | $\mathbf{0.7012}_{\pm 0.0045}$ | $0.2465_{\pm 0.0020}$ |

## D.3 COMPARISON TO HAMGNN AND HANG

In this section, we discuss on the differences with related works in the line of Hamiltonian systems for DGNs, such as HamGNN (Kang et al., 2023) and HANG (Zhao et al., 2023).

At first, we highlight that HamGNN and HANG are both instances of auto-differentiated models that are solved with a neuralODE solver, while our PH-DGN provides exact equations for the layer-wise

information aggregation, revealing a generalized formulation that allows to turn any MPNN-based model into its energy-preserving version, thus remaining efficient and general.

From an architectural point of view, HamGNN uses black-box Hamiltonian functions that are agnostic to the graph structure, thus limiting its interpretation as a DGN, while HANG explicitly couples the graph structure with the energy function, without making the connection nor investigate this design choice in relation to (long-range) message passing propagation, which is the main focus of our work. In contrast, our PH-DGN uses a port-Hamiltonian function that has a clear interpretation and is directly translated into the nonlinear message-passing framework, aiming to preserve information in the evolution and aggregation process. Thus, the port-Hamiltonian function drives the aggregation process based on the graph topology, allowing information to be propagated for an arbitrary number of layers. Additionally, our approach extends and generalizes Hamiltonian-inspired DGNs by introducing the ability to adaptively balance between conservative and dissipative behaviors, thus incorporating port-Hamiltonian dynamics. As demonstrated theoretically in Section 2 and empirically in Section 3, PH-DGN achieves long-range propagation through intrinsic graph coupling combined with conservative dynamics, which can be modulated by learnable components from the port-Hamiltonian framework, such as internal damping and external forces. These properties, crucial for adding generality to the approach and for effective information flow, are absent in existing models like HamGNN and HANG. We summarize the above comparison in Table 6 and in Table 7 we report an empirical assessment between PH-DGN, HamGNN, and HANG to evaluate the computational requirements of the three methods on the Eccentricity task (Section 3.3). To ensure a fair evaluation, we conducted a model selection given officially released implementations of HamGNN[1] and HANG[2] on a grid of total number of layers $L \in \{1, 5, 10, 20, 30\}$, embedding dimension $d \in \{20, 30\}$, step size $\epsilon \in \{1.0, 0.5, 0.1\}$, and in case of HamGNN we selected the Hamiltonian fuction to be $H \in \{H_2, H_3\}$, which are the top performing Hamiltonian functions in Kang et al. (2023). Note that for HamGNN, $L = 30$ was infeasible due to memory constraints of 40GB. Our results in Table 7 show that our PH-DGN not only outperforms HamGNN and HANG, but also operates at a lower computational cost, proving to be more efficient in terms of both speed and memory usage.

Table 6: Key properties of DE-DGN models inspired by Hamiltonian and port-Hamiltonian dynamics.

| | Hamiltonian Conservation | Graph Coupling | Learnable Driving Forces |
|---|---|---|---|
| **Ham.-insp. DGNs** | | | |
| HamGNN | ✓ | | |
| HANG | ✓ | ✓ | |
| **Our** | | | |
| PH-DGN$_C$ | ✓ | ✓ | |
| PH-DGN | | ✓ | ✓ |

Table 7: Mean test $log_{10}$(MSE) and std average over 4 training seeds on the Eccentricity task from Section 3.3 along with the measured memory consumption and average time per epoch (in seconds) of HamGNN, HANG, and our PH-DGN with $L = 20$ and $d = 30$. **Best** result is color coded.

| Model | Eccentricity $log_{10}$(MSE) ↓ | Memory GB ↓ | Time per epoch sec. ↓ |
|---|---|---|---|
| **Ham.-insp. DGNs** | | | |
| HamGNN | $0.7851_{\pm 0.0140}$ | 26.8 GB | $1773.43_{\pm 54.37}$ |
| HANG | $0.8302_{\pm 0.0051}$ | 3.2 GB | $34.63_{\pm 0.96}$ |
| **Our** | | | |
| PH-DGN | $\mathbf{-0.9348}_{\pm 0.2097}$ | **1.3 GB** | $\mathbf{17.13}_{\pm 0.06}$ |

---

[1] https://github.com/zknus/Hamiltonian-GNN
[2] https://github.com/zknus/NeurIPS-2023-HANG-Robustness

## D.4 ABLATION ON DRIVING FORCES

In order to study the different dampening and external forces we designed for the full port-Hamiltonian setup (i.e., including driving forces), we provide an ablation of our model on the Minesweeper task (Platonov et al., 2023), employing the most recent training protocol of Luo et al. (2024). The model selection is split into two stages. First, the best purely conservative PH-DGN (i.e., PH-DGN$_C$) is selected from a grid with total number of layers $L \in \{1, 5, 8\}$, embedding dimension $d \in \{256, 512\}$, step size $\epsilon \in \{0.1, 1.0\}$ according to the validation ROC-AUC. Then, the best selected configuration is tested with different driving forces. We refer the reader to Appendix A.2 for the details on the tested architectures. We report in Table 8 the results on the Minesweeper task alongside with the top-6 models from Luo et al. (2024). We observe that our purely conservative PH-DGN$_C$ show a strong improvement with respect to baseline methods, achieving new state-of-the-art performance. As evidenced by our ablation in Table 8, the inclusion of driving forces could not improve on this task, thus there is no advantage in deviating from a purely conservative regime. Our intuition is that counting-based tasks, like Minesweeper, greatly benefit from a purely conservative bias, such as that used by PH-DGN$_C$ since they require preserving all the information.

To complete the picture of the empirical performance differences between PH-DGN with and without driving forces, we report in Fig. 5 the ablation on the Graph Transfer task.
We observe that the purely conservative PH-DGN$_C$ achieves better performance than the full PH-DGN on two out of three tasks in the long-range regime. This finding aligns with our intuition that a purely conservative bias is essential for problems requiring the preservation of all information. Indeed, solving the graph transfer task effectively demands retaining the source node label without dissipation.

These results, together with those reported in Section 3, suggest that model selection should determine the optimal configuration of driving forces depending on the specific data setting.

Table 8: Ablation results on Minesweeper for different architectures of the driving forces proposed in this work. We report ROC-AUC scores averaged over the 10 official tr/vl/ts splits of Platonov et al. (2023). The **first**, **second**, and **third** best scores are colored.

| Model | | Train Score ROC-AUC ↑ | Val Score ROC-AUC ↑ | Test Score ROC-AUC ↑ |
|---|---|---|---|---|
| **Top-6 models from Luo et al. (2024)** | | | | |
| GraphGPS | | - | - | $90.75_{\pm 0.89}$ |
| SGFormer | | - | - | $91.42_{\pm 0.41}$ |
| Polynormer | | - | - | $97.49_{\pm 0.48}$ |
| GAT | | - | - | $97.73_{\pm 0.73}$ |
| GraphSAGE | | - | - | $97.77_{\pm 0.62}$ |
| GCN | | - | - | $\mathbf{97.86}_{\pm 0.24}$ |
| **Our - no driving forces** | | | | |
| PH-DGN$_C$ | | $99.78_{\pm 0.05}$ | $98.37_{\pm 0.22}$ | $98.45_{\pm 0.21}$ |
| **Our - with driving forces** | | | | |
| PH-DGN | | | | |
| **Dampening** | **External Force** | | | |
| – | MLP4-Sin | $99.37_{\pm 0.38}$ | $96.66_{\pm 0.35}$ | $96.61_{\pm 0.57}$ |
| – | DGN-tanh | $99.28_{\pm 0.10}$ | $97.29_{\pm 0.28}$ | $97.20_{\pm 0.42}$ |
| param | – | $99.79_{\pm 0.05}$ | $98.36_{\pm 0.22}$ | $98.42_{\pm 0.21}$ |
| param | MLP4-Sin | $99.55_{\pm 0.21}$ | $96.86_{\pm 0.33}$ | $96.86_{\pm 0.52}$ |
| param | DGN-tanh | $99.30_{\pm 0.19}$ | $97.35_{\pm 0.19}$ | $97.27_{\pm 0.29}$ |
| MLP4-ReLU | – | $99.62_{\pm 0.57}$ | $95.33_{\pm 0.36}$ | $95.33_{\pm 0.65}$ |
| MLP4-ReLU | MLP4-Sin | $99.93_{\pm 0.03}$ | $95.79_{\pm 0.57}$ | $95.67_{\pm 0.62}$ |
| MLP4-ReLU | DGN-tanh | $98.79_{\pm 0.24}$ | $95.42_{\pm 0.45}$ | $95.41_{\pm 0.66}$ |
| DGN-ReLU | – | $94.96_{\pm 0.17}$ | $93.22_{\pm 0.81}$ | $93.42_{\pm 0.61}$ |
| DGN-ReLU | MLP4-Sin | $95.61_{\pm 0.48}$ | $93.80_{\pm 0.71}$ | $93.87_{\pm 0.55}$ |
| DGN-ReLU | DGN-tanh | $95.01_{\pm 0.47}$ | $93.21_{\pm 0.72}$ | $93.32_{\pm 0.84}$ |

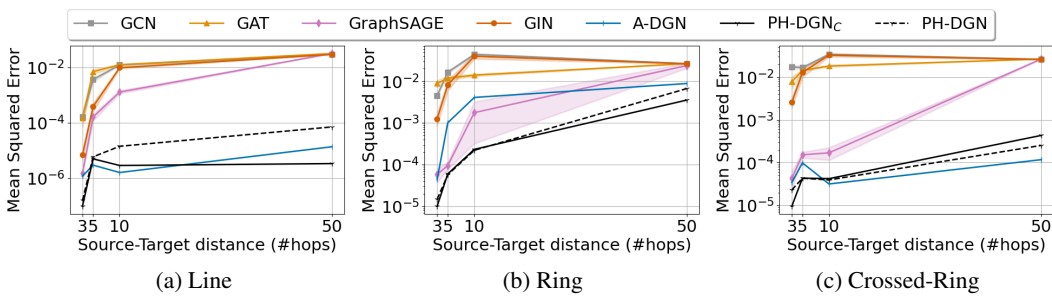

Figure 5: Information transfer performance on (a) Line, (b) Ring, and (c) Crossed-Ring graphs, including the purely conservative PH-DGN (i.e., PH-DGN$_C$) and PH-DGN with driving forces.

