# OpenReview forum: "Port-Hamiltonian Architectural Bias for Long-Range Propagation in Deep Graph Networks"
_ICLR.cc/2025/Conference — ICLR 2025 Poster_

### Official Review · Reviewer_7PXw · 2024-10-25

**Soundness:** 4
**Presentation:** 3
**Contribution:** 3
**Rating:** 8
**Confidence:** 3

**Summary:**

This work provides a novel methodology for the incorporation of port-Hamiltonian dynamics in graph representation learning. The central model, called a port-Hamiltonian Deep Graph Network (PH-DGN), is introduced first in a purely conservative setting using only the Hamiltonian and no non-conservative terms. Several theorems are developed to show that this conservative case leads to long-range data propagation between graph nodes, where the graph dynamics exhibit no energy loss and gradients do not vanish as the backward sensitivity matrix is bounded below. Dissipitive forces are then added to the full port-Hamiltonian model, which are two additional networks that may be trained to capture non-conservative forces. Several experiments follow, including a showcase of energy conservation and sensitivity to empirically verify theoretical work, and a graph transfer problem and graph property prediction problem to compare performance on benchmark tasks against other graph models in settings which require long-range information propagation.

**Strengths:**

- There is very clear motivation to this work, and it builds nicely upon other references.
- The proposed port-Hamiltonian approach is an original and clever way to allow for non-conservative dynamics in a graph network while still maintaining long-range message passing.
- The theoretical results for the conservative case are strong, and the motivation and interpretation for these are presented nicely.
- The experimental setup is superb; the care taken to ensure ease of replication is applauded. Model details are presented very clearly and choices are explained well for each step of the setup.
- A strong suite of models are compared against, with many different competitors and different approaches. The consistently favorable results provide a great deal of strength to claims of the proposed method's performance.
- The appendices are comprehensive for both proofs and experimental setup, and made clear many of the questions I had on an initial read.

**Weaknesses:**

- The majority of the theoretical results are developed for the conservative case. This makes sense in the context, as conservative long-range message passing is stated as a goal, but I would also be quite interested to see what could be proven for the fully general port-Hamiltonian case.
- In the explanation of Theorem 2.3, the statement that "the final representation of each node retains its complete past" seems somewhat strong. While I understand that the BSM result shows the influence of the entire past history on the current state, this statement as written seems to imply something stronger, and perhaps could be made more clear.
- The dissipitive force terms are added in some experiments to great success, but the explanations of their performance are more intuitive and are not supported by hard data in the paper. There may be a great opportunity here for visualization to support the intuitive claims.

There are two very minor typos:
- In the first paragraph of Section 2, "node states' in graph" has an unnecessary apostrophe.
- In Appendix D.3, "on a grid of n. layers" has an unnecessary period.

**Questions:**

- How would classical GCN aggregation interact with Theorem 2.4? Can the bound be easily extended for that case?
- In Section 3.1, you mention that the growing behavior can be controlled by regularizing the weight matrices or using normalized aggregation functions. Did you try this? How are the empirical results?
- Have you examined the interpretability of a trained PH-DGN? In particular, do the learned non-conservative forces make sense for the associated problem?

---

> ### Author Response · Authors · 2024-11-19
> **Rebuttal Part 1**
>
> We sincerely thank the Reviewer for the consistently positive feedback and the valuable insights provided on our work. In particular, we appreciate the Reviewer’s recognition of the ***very clear motivation*** behind our approach, as well as its ***originality*** and ***cleverness***. Additionally, we are grateful for the acknowledgment of our ***strong theoretical results***, our ***superb experimental setup***, and the ***consistently favorable results*** demonstrated. We are pleased to address each comment in detail, following the original order.
>
> **Regarding theoretical insights in the full port-Hamiltonian case**
>
> We appreciate the Reviewer’s effort in improving the quality of our work. In this regard, we included the Reviewer’s suggestion in Appendix B.7, where we derived new additional theoretical results on the effects of the port-Hamiltonian components on information propagation. Specifically, we observe that the sensitivity matrix can be linearly decomposed into the conservative and dissipative terms. Under the assumption that the driving forces and their derivatives are bounded, the self-influence of a node after one update can be constrained within fixed upper and lower bounds, which are mediated (among the other) by the step-size $\epsilon$. Additionally, we demonstrate that a similar upper bound applies to the influence between neighboring nodes. These results indicate that, under mild assumptions, the port-Hamiltonian components theoretically support long-range propagation.
>
> **Regarding the explanation of Theorem 2.3**
>
> We thank the Reviewer for carefully reading our manuscript and helping us to further improve on the presentation of our work. We acknowledge that the wording “retains its complete past” may be ambiguous and leaves room for interpretation about how “strong” the past influences the final representation. However, since the sensitivity matrix does not vanish, we believe that the information can not be discarded. Moreover, as our upper bound on the self-node BSM in Appendix A.1 shows, the further the node representation lies in the past, the higher is its influence on the final state to achieve the energy conservation regime. We are happy to further discuss with the Reviewer on this regard, since we do not fully grasp the intended meaning of the  “imply something stronger” comment.
>
> **Regarding the explanations of the performance gains coming from driving forces**
>
> We thank the Reviewer again for pointing us to opportunities to further improve our work. Dampening and external forces have great interpretations stemming from physical phenomena, like a pendulum that is suffering friction losses but at the same time is externally moved around in space. In our scenario, we believe that the learned driving forces act as an adaptive filter mechanism that filters noisy information in the embedding evolution, facilitating the learning of relevant information, thus resulting in improved performance on the downstream task. However, visualizing the filtering mechanism poses a significant challenge because we don’t have explicit knowledge of what constitutes "relevant" versus "noisy" information. Similarly, visualizing the high-dimensional node dynamic trajectories is also not trivial, since the 2D representation may not reflect the actual non-linear behavior in the latent space.
>
> **Regarding minor typos**
>
> We thank the Reviewer for thorough reading of our work. We corrected them in the revised paper.
>
> **Regarding classical GCN in Theorem 2.4**
>
> We thank the Reviewer for the comment, since our goal is to provide a general framework which can be used with many different aggregation schemes. Considering the difference between our vanilla neighborhood aggregation and the degree-normalized GCN sum, we find that the constants $\frac{1}{\sqrt{d_u}\sqrt{d_v}}$ can be injected as part of a weighted adjacency matrix into our proof. For the ease of presentation, we opt for omitting these kinds of scaling factors in the proof.

---

> ### Author Response · Authors · 2024-11-19
> **Rebuttal Part 2**
>
> **Regarding controlling the upper bound Appendix Th. A.1**
>
> We thank the Reviewer for the question. In our experiments, we tested two aggregations schemes, i.e., the one implemented in Eq. 6 and the classical GCN aggregation. Although Theorem A.1 theoretically indicates a potential increase in the sensitivity measure, we did not observe this issue with either aggregation method during our experiments. Furthermore, we found that incorporating the GCN aggregation scheme did not consistently lead to improved performance. Hence, we did not see the necessity to include norm-constraining regularizers on the weights during training in our experiments. We recommend practitioners to treat the aggregation scheme as a hyperparameter to be selected via model selection.
>
> **Regarding the interpretability of a trained PH-DGN**
>
> We thank the Reviewer for the comment. Our experimental suite was designed to deliberately demonstrate the long-range capabilities of our approach and, as such, we did not focus on examining the interpretability of the trained model. Interpreting the dynamics of the model is inherently challenging due to the lack of ground truth for what constitutes the "true" flow of information across the graph, especially on tasks like predicting 3D properties of peptides. Without this reference, it becomes difficult to disentangle how the model processes and propagates information or to verify whether it aligns with any hypothesized dynamics. This challenge is further amplified by the fact that, in our experiments, the driving forces are modeled using complex neural networks, making it difficult to deduce clear intuitions about how these forces operate within the model. These limitations highlight the need for future work to explore the interpretability of the methods.

---

> > ### Comment · Reviewer_7PXw · 2024-11-27
> > **Rebuttal Response**
> >
> > Thank you for the thorough rebuttal and explanations! They increased my understanding of the work.

---

### Official Review · Reviewer_XxSd · 2024-11-03

**Soundness:** 2
**Presentation:** 3
**Contribution:** 2
**Rating:** 5
**Confidence:** 2

**Summary:**

This paper proposes PH-DGN, a new GNN based on the port-Hamiltonian system to develop GNN that can solve graph learning tasks that require long-range dependencies. Two variants of PH-DGN are proposed: a conservative PH-DGN based on the Hamiltonian system and a PH-DGN based on the port-Hamiltonian system by introducing learnable dissipative terms.
The theoretical analyses show that the conservative PH-DGN is stable and energy-preserving as a dynamical system, and derive a lower bound for the sensitivity, implying the possibility of long-range interaction.
Numerical experiments show that the conservative PH-DGN is energy-preserving without gradient vanishing using synthesis datasets (Section 3.1) and that the long-range interaction can be achieved in graph learning tasks that require long-range interactions (Sections 3.2, 3.3). Also, the usefulness of two variants of PH-DGN is evaluated on long-range graph benchmarks of real datasets.

**Strengths:**

- The background knowledge of the port-Hamiltonian system is explained carefully, increasing accessibility for readers unfamiliar with this topic.
- For conservative PH-DGN, the ability of long-range interaction is theoretically shown by giving the lower bounds of sensitivity.
- The validity of the proposed methods for synthesis datasets is properly demonstrated.

**Weaknesses:**

- As the title indicates, the theme of this paper is the relationship between GNN and long-range propagation based on the port-Hamiltonian system. However, no theoretical guarantees on long-range propagation are given for general PH-DGNs with dissipative components.
- The tables of experimental results could be clearer (in particular Tables 2 and 5).
- For $\mathrm{PH-GDN}_{\mathrm{C}}$, which has theoretical guarantees, the prediction performance on the real dataset (long-range graph benchmark) does not outperform existing methods, and hence its practical usefulness is limited.

**Questions:**

**Questions**

- l.161: What is the definition of anti-derivative of an activation function $\sigma$?
- l.223, l.228: Which norm is used in $\| \partial \mathbf{b}_u(T) / \partial \mathbf{b}_u(T-t) \|$?
- l.259: *Therefore, [...]. , it holds the capability of PH-GDN to perform long-range propagation effectively*: the authors compare the upper bounds of sensitivity for the existing model (Theorem 2.3) and the proposed model (Theorem 2.4), then argue that since the latter is larger than the former, the proposed model is more effective in performing long-range propagation. However, it is difficult to claim so because there is the possibility that Theorem 2.4 only shows a looser bound than Theorem 2.3, which does not exclude the possibility that the upper bound does not properly reflect the sensitivity of the PH-GDN. It should be shown theoretically or experimentally that this upper bound adequately reflects the sensitivity of the PH-GDN.
- l.1129: I want to clarify the setup of the Graph Transfer Task: If I understand correctly, the feature vectors of all nodes are 1-dimensional and randomly sampled from $\mathrm{Unif}([0, 0.5))$. The target value is 0 for the source node and 1 for the target node. Assuming that this problem setup is correct, I need help understanding why this task is solvable because the model cannot distinguish source or target nodes from other nodes from feature vectors.


**Minor Comments**

- l.196: *non-dissipative (i.e., long-range)*: I think this is a slightly misleading expression. My understanding is that this paper uses non-dissipative in the sense of energy-perserving. Although non-dissipative implies long-range propagation (Theorem 2.3), they are not equivalent. In fact, this paper argues that non-dissipative PH-DGN performs better than conservative PH-DGN in predicting the LRGB dataset in some numerical experiments.
- l.436: The use of position encoding is only mentioned in the caption of Table 2 and should be explicitly stated in the text, specifically in the setup of Section 3.4.
- l.436: The correspondence between Table 2 and Table 5 needs to be clarified. For example, the method titled *MPNNs* in Table 2 is titled *re-evaluated* in Table 5. However, GCNII is not listed as *re-evaluated*, but as *MPNNs*. Furthermore, it is difficult to tell from the captions of Table 5 whether the authors test each listed method by themselves or is a citation of existing research. The reference should be indicated for each method in Table 5, for example, by adding a column such as *reference* to Table 5.
- l.443: *Overall, our port-Hamiltonian framework [...] shows great benefit [...] without requiring additional strategies such as global position encoding, global attention mechanism, or rewiring techniques [...].*: I suggest clarifying how the usefulness of each strategy is shown by comparing it with existing methods. More specifically:
  - The superiority of the proposed method over global position encoding is justified by the comparison with MPNN-based models using position encoding.
  - The superiority over the global attention mechanism is by comparison with the Transformer-based method.
  - The superiority of rewiring methods by comparison with Drew.
- l.1153: The reference to Adam (Kingma & Ba, 2015) should be cited.
- l.1343: n. layers -> the number of layers
- l.1353: Table 6 claims that PH-DGN achieves both Hamiltonian conservation and Learnable driving forces. However, I think this is misleading because to achieve Hamiltonian conservation, the dissipative component must be removed, resulting in $\mathrm{PH-DGN}_{\mathrm{C}}$. It is not possible to achieve both characteristics at the same time in one architecture. Instead, two variants that achieve only one of the two are proposed.

**Details Of Ethics Concerns:**

N.A.

---

> ### Author Response · Authors · 2024-11-19
> **Rebuttal Part 1**
>
> We thank the Reviewer for the extensive feedback on our manuscript and for acknowledging that we  ***carefully explained*** the knowledge behind our method and that we ***theoretically show*** and empirically validate the ***usefulness*** in the long-range regime.  Below, we address each of your comments, for which we are grateful. We found them to be helpful to further improve the quality of our paper, and we hope that you are satisfied with our response. We hope that in light of our clarifications and modifications to the paper, you will consider revising your score.
> In particular, following the Reviewer’s suggestions, we highlight how the revised version of the paper now contains new additional theoretical guarantees on long-range propagation that also accounts for the presence of non-conservative forces, as well as new experiments to prove the usefulness of the fully conservative PH-DGN (and, of course, additional clarifications to all Reviewer’s questions).
>
> **1) Regarding the theoretical guarantees for the general PH-DGN**
>
> We appreciate the Reviewer’s effort in improving the quality of our work. We note that the main goal of our paper is to reconcile under a single framework strong theoretical guarantees of conservation, for non-dissipative long-range propagation, with non-conservative behaviors, to potentially improve the performance on the downstream task. Indeed, without driving forces, the final ability of the system to model all complex nonlinear dynamics is restricted in real-world scenarios, as empirically shown in Table 2. To further accommodate the Reviewer’s suggestion, we derived some additional theoretical results on the effects of the port-Hamiltonian components on information propagation in Appendix B.7. In particular, we note that the sensitivity matrix can be linearly decomposed into the conservative and dissipative terms. Assuming that the driving forces and their derivatives are bounded, the self-influence of a node after one update can be constrained within fixed upper and lower bounds which are mediated (among the other) by the step-size $\epsilon$. Additionally, we demonstrate that a similar upper bound applies to the influence between neighboring nodes. These results indicate that, under mild assumptions, the port-Hamiltonian components theoretically support long-range propagation.
>
>
> **2) Regarding Tables 2 and 5**
>
> We thank the Reviewer for the feedback. Both Tables 2 and 5 contain results for the LRGB benchmark. Due to submission length constraints, we decided to report in the main text (i.e., Table 2) only a selection of all the considered baselines while in the appendix (i.e., Table 5) we report the full list of baselines. In our revised paper, we have improved the clarity of both tables by better specifying which result comes from which paper, and whether or not the model uses positional/structural encodings. We are open to other suggestions to improve the clarity of the tables.

---

> ### Author Response · Authors · 2024-11-19
> **Rebuttal Part 2**
>
> **3) Regarding the usefulness of PH-DGN$_C$ on the real dataset**
>
> We appreciate the Reviewer’s comment. First, we would like to highlight that, given our PH-DGN is designed through the lens of differential equations, the most relevant baselines for comparison are DE-DGN models. As shown in Table 2, PH-DGN$_C$   achieves superior performance compared to all DE-DGN baselines. Furthermore, PH-DGN$_C$ outperforms global graph transformers and multi-hop approaches across both tasks, and ranks as the third-best overall model on the peptide-func dataset.
> Overall, PH-DGN$_C$ outperforms 12 out of 13 baselines on peptide-func and 7 out of 13 baselines on the other task.
> In general, our intuition is that PH-DGN$_C$  may have improved utility for tasks that require preserving ***all*** information, such as propagating all node labels over the graph or counting the number of nodes with a given label. To provide empirical support for this intuition, we have included in the revised manuscript (Appendix D.4) an ablation study on the Minesweeper task from (Platonov et al. (2023)) and on the graph transfer task. For your convenience, we report the Minesweeper results in the table below.
> In the Minesweeper task (i.e., a counting-based task), PH-DGN shows no advantage in deviating from a purely conservative regime. Notably, our PH-DGN$_C$ achieves a new state-of-the-art performance of 98.45 on this benchmark. These results suggest that model selection should determine the optimal use of port components depending on the specific data setting.
>
>
> | **Model**                       	| **Train Score (ROC-AUC ↑)**	| **Test Score (ROC-AUC ↑)**	|
> |-------------------------------------|--------------------------------|-------------------------------|
> | Top-6 models form Luo et al. (2024) | | |
> | GraphGPS | - | 90.75 ± 0.89 |
> | SGFormer |  - | 91.42 ± 0.41 |
> | Polynormer | - | 97.49 ± 0.48 |
> | GAT | - | 97.73 ± 0.73 |
> | GraphSAGE                      	| -                          	| 0.9777 ± 0.0062          	|
> | GCN                              	| -                          	| **0.9786 ± 0.0024**      	|
> | **Our - no driving forces**    	|                            	|                           	|
> | PH-DGN$_C$         	| 0.9978 ± 0.0005            	| **0.9845 ± 0.0021**      	|
> | **Our - with driving forces**  	|                            	|                           	|
> | *PH-DGN*                       	|                            	|                           	|
> | *Dampening* / *External Force* 	|                            	|                           	|
> | -- / MLP4-Sin                  	| 0.9937 ± 0.0038            	| 0.9661 ± 0.0057          	|
> | -- / DGN-tanh                  	| 0.9928 ± 0.0010            	| 0.9720 ± 0.0042          	|
> | param / --                     	| 0.9979 ± 0.0005            	| **0.9842 ± 0.0021**      	|
> | param / MLP4-Sin               	| 0.9955 ± 0.0021            	| 0.9686 ± 0.0052          	|
> | param / DGN-tanh               	| 0.9930 ± 0.0019            	| 0.9727 ± 0.0029          	|
> | MLP4-ReLU / --                 	| 0.9962 ± 0.0057            	| 0.9533 ± 0.0065          	|
> | MLP4-ReLU / MLP4-Sin           	| 0.9993 ± 0.0003            	| 0.9567 ± 0.0064          	|
> | MLP4-ReLU / DGN-tanh           	| 0.9789 ± 0.0024            	| 0.9541 ± 0.0066          	|
> | DGN-ReLU / --                  	| 0.9496 ± 0.0017            	| 0.9342 ± 0.0061          	|
> | DGN-ReLU / MLP4-Sin            	| 0.9561 ± 0.0048            	| 0.9387 ± 0.0055          	|
> | DGN-ReLU / DGN-tanh            	| 0.9501 ± 0.0047            	| 0.9332 ± 0.0084          	|
>
> Platonov et al. A critical look at the evaluation of GNNs under heterophily: Are we really making progress?. ICLR 2023
>
> Now we are happy to clarify on each question, following original ordering:
>
> **- Regarding the term “anti-derivative”**
>
> We use the term “anti-derivative” to refer to a differentiable function $F$ whose derivative is equal to the original function $f$. Therefore, the anti-derivative of an activation function is the function $F(x)$ whose derivative leads to the original activation function, i.e.,  $F^\prime(x) = \sigma (x)$.
>
> **- Regarding the norm of the sensitivity**
>
> Since $[\partial x_u(T)/ \partial x_u(T-t)]$ is an $\mathbb{R}^{d\times d}$ matrix, our statements are valid for all sub-multiplicative matrix norms, e.g., p-norm and Frobenius norm

---

> ### Author Response · Authors · 2024-11-19
> **Rebuttal Part 3**
>
> **- Regarding upper bound sensitivity**
>
> We thank the Reviewer for the insightful comment. To clarify, Theorem 2.3 provides a lower bound for the influence of the same node $u$, while Theorem 2.4 establishes an upper bound for the influence between two different nodes, $u$ and $v$, both derived using our proposed model. The study of upper bounds for interactions between different nodes was recently proposed by Topping et al. (2022) and Di Giovanni et al. (2023) as a means to characterize the long-range propagation problem and it is currently a widely accepted means to characterize the long-range propagation problem within the community of deep learning for graphs. We agree with the Reviewer that an upper bound may not fully reflect the actual long-range capability of the model. However, we believe that, since information cannot vanish and our PH-DGN$_C$ can theoretically propagate more information, our PH-DGN$_C$ is theoretically more effective in long-range propagation. This claim is supported by our experimental results presented in Section 3. Specifically, our model (with the bigger upper bound) achieves better results on ***all*** long-range tasks than MPNN models (with a smaller upper bound, as shown in Topping et al. 2022 and Di Giovanni et al. 2023). Therefore, we believe that our theoretical bounds reflect the practical sensitivity of PH-DGN, demonstrating its superior capability to propagate information over long ranges compared to previous state-of-the-art models.
>
> Topping et al. Understanding over-squashing and bottlenecks on graphs via curvature. ICLR 2022
>
> Di Giovanni et al. On over-squashing in message passing neural networks: the impact of width, depth, and topology. ICML 2023
>
> **- Regarding the graph transfer task**
>
> We refer the Reviewer to Appendix C.2 for deeper insights on the data and the task of the graph transfer experiment. We built this experiment based on the graph transfer task proposed by Di Giovanni et al. (2023). In simple words, it is an information-exchange task where we measure how far the information can travel within the graph. The source node is assigned with feature “1”, the target with feature “0”, and intermediate nodes with a random feature sampled from a uniform distribution in [0, 0.5). Then, we implemented a supervised task  (specifically, a regression problem) that measures how much of the source node information has reached the target node. On large graphs, the more information reaches the target node, the more effective the model is in the long-range regime. We clarified this in the revised manuscript.
>
> Di Giovanni et al. On over-squashing in message passing neural networks: the impact of width, depth, and topology. ICML 2023
>
> We now move to the minor comments, following original ordering. Again, we thank the Reviewer for working with us in improving the quality of our work.
>
> **- Regarding non-dissipativeness**
>
> The Reviewer is correct that we designed our PH-DGN as a non-dissipative system in the sense of “energy-preserving”. Specifically, we built our PH-DGN on top of recent literature on non-dissipative dynamical systems (Haber et al. (2017); Bo et al. (2019); Gravina et al. (2023)), which show that a non-dissipative behavior allows capturing long-term (in the case of time series) and long-range (in the graph domain) dependencies. In a broader sense, the energy of the system can be associated to the node information, since it is linked to the node embedding sensitivity. We believe that our PH-DGN with driving forces achieves better performance on the real-world LRGB because the driving forces act as an adaptive filter mechanism that filters noisy information, facilitating the learning of relevant information. As discussed in the newly introduced Appendix D.4, there are scenarios where a purely non-dissipative approach is more beneficial, as it ensures no loss of information. Although we believe there is a strong correlation between long-range and non-dissipative, we acknowledge the source of misunderstanding and rephrased the sentence in the revised draft.
>
> Haber et al. Stable architectures for deep neural networks. Invers Problems 2017
>
> Bo et al. AntisymmetricRNN: A dynamical system view on recurrent neural networks. ICLR 2019
>
> Gravina et al. Anti-Symmetric DGN: a stable architecture for Deep Graph Networks. ICLR 2023
>
> **- Regarding positional encoding in the text**
>
> The positional/structural encoding refers only to MPNN baselines, while our model does not rely on such an approach. We clarified this aspect in the revised manuscript (Table 2 and Section 3.4).
>
> **- Regarding Tables 2 and 5**
>
> We thank the Reviewer for the effort in improving the quality of our work.  We clarified this aspect in the revised manuscript.

---

> ### Author Response · Authors · 2024-11-19
> **Rebuttal Part 4**
>
> **- Regarding the comparison clarification**
>
> Again, we thank the Reviewer for its effort and we refer them to the revised manuscript, which now contains a deeper clarification on the usefulness of the proposed model with respect to existing methods.
>
> **- Regarding Adam citation**
>
>  We included the citations to our employed optimization strategies in the revised manuscript
>
> **- Regarding n. layer typo**
>
> We corrected the typo in the revised manuscript. Thank you.
>
> **- Regarding Table 6**
>
> We thank the Reviewer for the comment. Appendix Table 6 serves as a high-level comparison with related works on Hamiltonian inspired DGNs. We opted to report our framework as a single row in the table for simplicity reasons, since in our PH-DGN the driving forces can be turned on and off depending on the specific needs of the problem. Indeed, from a high-level perspective, the conservative approach could be seen as a subset of the full port-Hamiltonian approach, explaining why in the single row scenario we marked the Hamiltonian conservation column. Following the Reviewer’s suggestion, we revised Table 6 to clarify that driving forces do not allow for purely Hamiltonian conservation.

---

> ### Comment · Reviewer_XxSd · 2024-11-24
> **Response to authors' rebuttal (1/3)**
>
> **1) Regarding the theoretical guarantees for the general PH-DGN**
>
> > To further accommodate the Reviewer's suggestion, we derived some additional theoretical results on the effects of the port-Hamiltonian components on information propagation in Appendix B.7. [...] These results indicate that, under mild assumptions, the port-Hamiltonian components theoretically support long-range propagation.
>
> I thank the authors for providing additional theoretical results on PH-DGNs with dissipative components. From Theorem B.1--B.3, I understand that when we introduce the small dissipative components, the perturbation of the BSM is small as well, guaranteeing the non-vanishing gradient and long-range propagation.
>
>
> **2) Regarding Tables 2 and 5**
>
> > In our revised paper, we have improved the clarity of both tables by better specifying which result comes from which paper, and whether or not the model uses positional/structural encodings.
>
> I thank the authors for updating the tables. The update improves their clarity. I have several clarifications:
>
> 1. Does **Re-evaluated** in Table 5 mean that the results of GCNII+PE/SE and DRew-GCN+PE/SE are re-evaluated by the authors and not cited from Tönshoff et al (2023)?
> 2. If 1 is true, could you let me know why the authors re-evaluated these models?
>
> In addition, I have the following suggestions for further improvement. However, since this is a matter of taste, it is OK that the authos opts not to adopt them:
>
> 1. While the order of Table 2 is Tönshoffetal -> Multi-hop (Gutteridge) -> Transformers (Gutteridge), that of Table 5 is Multi-hop (Gutteridge) -> Transformers (Gutteridge) -> Tönshoff. Tables 2 and 5 should have the same order of sections.
> 2. I think the section title of Tönshoffetal in Table 5 should be **MPNNs** rather than **Re-evaluated** because not all models are re-evaluated and other section titles are model names (e.g., Multi-hop DGNs)
>
>
> **3) Regarding the usefulness of PH-DGN$_C$ on the real dataset**
>
> > For $\mathrm{PH-GDN}_{\mathrm{C}}$, which has theoretical guarantees, the prediction performance on the real dataset (long-range graph benchmark) does not outperform existing methods, and hence its practical usefulness is limited.
>
> I thank the authors for responding to my concerns about the practical usefulness of $\mathrm{PH-GDN}_{\mathrm{C}}$
>
> > We appreciate the Reviewer's comment. First, we would like to highlight that, given our PH-DGN is designed through the lens of differential equations, the most relevant baselines for comparison are DE-DGN models.
>
> I think the design of the numerical experiment should reflect the messages the authors want to claim. From the practitioners' point of view, one of the most important criteria for selecting models is prediction performances. For them, the origin of the model architecture, especially whether the architecture is derived from differential equations, is not important. If the authors emphasize the comparison between PH-DGN with other DE-DGN models, I want to clarify what what the authors want to claim by this comparison.
>
> > Furthermore, PH-DGN$_C$ outperforms global graph transformers and multi-hop approaches across both tasks and ranks as the third-best overall model on the peptide-func dataset.
> > Overall, PH-DGN$_C$ outperforms 12 out of 13 baselines on peptide-func and 7 out of 13 baselines on the other task.
>
> As in the previous discussion, I want to clarify what the authors want to claim by these comparisons.
>
> > To provide empirical support for this intuition, we have included in the revised manuscript (Appendix D.4) an ablation study on the Minesweeper task from (Platonov et al. (2023)) and on the graph transfer task.
>
> I want to clarify the experiment setting of this task, more specifically, the following part:
>
> > The model selection is split into two stages. First, the best purely conservative PH-DGN (i.e., PH-DGN$_C$) is selected from a grid with total number of layers L∈{1,5,8}, embedding dimension d∈{256,512}, step size ϵ∈{0.1,1.0} according to the validation ROC-AUC. Then, the best selected configuration is tested with different driving forces.
>
> Given training and validation datasets, I wonder how the learnable parameters in models for dampening and external forces are used for choosing the hyperparameters of PH-DGN$_C$.
>
> -------------------
>
> **Regarding the term "anti-derivative"**
>
> I thank the authors for the explanation. I did not know the primitive integral is also called the anti-derivative. I am sorry that I should have checked it by myself.
>
>
> **Regarding the norm of the sensitivity**
>
> > Since $[\partial x_u(T)/ \partial x_u(T-t)]$ is an $\mathbb{R}^{d\times d}$ matrix, our statements are valid for all sub-multiplicative matrix norms, e.g., p-norm and Frobenius norm
>
> I understand that the proof is valid for any sub-multiplicative norm. I suggest explicitly writing that the norm is any sub-multiplicative to the statement.

---

> > ### Comment · Reviewer_XxSd · 2024-11-24
> > **Response to authors' rebuttal (2/3)**
> >
> > **Regarding upper bound sensitivity**
> >
> > > [...] To clarify, Theorem 2.3 provides a lower bound for the influence of the same node $u$, while Theorem 2.4 establishes an upper bound for the influence between two different nodes, $u$ and $v$, both derived using our proposed model.
> >
> > I thank the authors for pointing it out. Since this question was based on my misunderstandings, I want to withdraw it.
> >
> >
> > > The study of upper bounds for interactions between different nodes was recently proposed by Topping et al. (2022) and Di Giovanni et al. (2023) as a means to characterize the long-range propagation problem and it is currently a widely accepted means to characterize the long-range propagation problem within the community of deep learning for graphs.
> >
> > I agree with the authors' arguments that assuming that the upper bound of theoretical guarantees reflects the properties of ML models is one of the standard (if not perfect) methods in ML theory, such as statistical learning theory.
> >
> > > This claim is supported by our experimental results presented in Section 3. Specifically, our model (with the bigger upper bound) achieves better results on all long-range tasks than MPNN models (with a smaller upper bound, as shown in Topping et al. 2022 and Di Giovanni et al. 2023).
> >
> > I also agree that the model's better performance in long-range tasks justifies the comparison using the upper bounds.
> >
> >
> > **Regarding the graph transfer task**
> >
> > > [...] The source node is assigned with feature "1", the target with feature "0", and intermediate nodes with a random feature sampled from a uniform distribution in $[0, 0.5)$. Then, we implemented a supervised task (specifically, a regression problem) that measures how much of the source node information has reached the target node. [...]
> >
> > I thank the authors for clarification of the problem setup. However, I still do not understand the setup. I guess I could not fully figure out what the authors assume implicitly. More specifically, I understand the feature vector is 1 for the source node, 0 for the target node, and random value from $[0, 0.5)$ for intermediate nodes. Then, how about the target value of the regression task? Do we assign a value to each node in a graph (i.e., node prediction task) or a single value to the whole graph (i.e., graph prediction task)?
> >
> >
> > **Regarding non-dissipativeness**
> >
> > > The Reviewer is correct that we designed our PH-DGN as a non-dissipative system in the sense of "energy-preserving". Specifically, we built our PH-DGN on top of recent literature on non-dissipative dynamical systems (Haber et al. (2017); Bo et al. (2019); Gravina et al. (2023)), which show that a non-dissipative behavior allows capturing long-term (in the case of time series) and long-range (in the graph domain) dependencies. In a broader sense, the energy of the system can be associated to the node information, since it is linked to the node embedding sensitivity. We believe that our PH-DGN with driving forces achieves better performance on the real-world LRGB because the driving forces act as an adaptive filter mechanism that filters noisy information, facilitating the learning of relevant information. As discussed in the newly introduced Appendix D.4, there are scenarios where a purely non-dissipative approach is more beneficial, as it ensures no loss of information.
> >
> > I thank the authors for the detailed explanation of the intuition about the relationship between the non-dissipative system and long-range dependencies.
> >
> > > Although we believe there is a strong correlation between long-range and non-dissipative, we acknowledge the source of misunderstanding and rephrased the sentence in the revised draft.
> >
> > I do not deny the strong correlation between the two concepts. Instead, I also think they are interdependent. However, whether they are equivalent is unknown; we need more studies to claim so. Therefore, I think it is better to treat these concepts differently.
> >
> >
> > **Regarding positional encoding in the text**
> >
> > > The positional/structural encoding refers only to MPNN baselines, while our model does not rely on such an approach. We clarified this aspect in the revised manuscript (Table 2 and Section 3.4).
> >
> > OK. I thank the authors for the clarification.
> >
> >
> > **Regarding Tables 2 and 5**
> >
> > > We thank the Reviewer for the effort in improving the quality of our work. We clarified this aspect in the revised manuscript.
> >
> > OK. See my response to **2) Regarding Tables 2 and 5** for my suggestions.
> >
> >
> > **Regarding the comparison clarification**
> >
> > > Again, we thank the Reviewer for its effort and we refer them to the revised manuscript, which now contains a deeper clarification on the usefulness of the proposed model with respect to existing methods.
> >
> > OK. I thank the authors for the clarification.

---

> > > ### Comment · Reviewer_XxSd · 2024-11-24
> > > **Response to authors' rebuttal (3/3)**
> > >
> > > **Regarding Adam citation**
> > >
> > > > We included the citations to our employed optimization strategies in the revised manuscript
> > >
> > > OK. I thank the authors for adding the reference.
> > >
> > >
> > > **Regarding n. layer typo**
> > >
> > > > We corrected the typo in the revised manuscript. Thank you.
> > >
> > > OK.
> > >
> > >
> > > **Regarding Table 6**
> > >
> > > > Following the Reviewer's suggestion, we revised Table 6 to clarify that driving forces do not allow for purely Hamiltonian conservation.
> > >
> > > OK. I thank the authors for considering my comments and updating the table.

---

> > > > ### Author Response · Authors · 2024-11-25
> > > > **Response to Reviewer's rebuttal**
> > > >
> > > > We thank the Reviewer for continuing to engage with us and for positively acknowledging the majority of our clarifying responses. Below, we provide our responses to the last open questions. We hope that these clarifications will help the Reviewer reconsider their overall assessment of our work.
> > > >
> > > > ---
> > > >
> > > > **2) Regarding Tables 2 and 5**
> > > >
> > > > We thank the Reviewer for acknowledging the improved clarity of our Tables 2 and 5. We are happy to further elaborate on the specific wording used in the Tables.
> > > > We employ the term “re-evaluated” to point to the re-evaluation performed by Tönshoff et al. (2023) with a different training protocol. Therefore, we used “re-evaluated” to reflect different performances of already presented models. We agree with the Reviewer that this term may be a reason of misunderstanding and hence we propose to rephrase it as “Modified experimental protocol, Tönshoff et al. (2023)”.
> > > >
> > > > The Reviewer is correct that we computed the scores for GCNII+PE/SE and DRew-GCN+PE/SE strickly following the experimetal setup of Tönshoff et al. (2023). The reason for this choice was (i) to further underscore the overall performance improvements introduced by our PH-DGN by providing the reader with a more complete picture, which includes the missing SOTA method in Tönshoff et al.'s re-evaluation; and (ii) to comply with this Reviewer's request to include GCNII in the "re-evaluated" section of the table.
> > > >
> > > > **3) Regarding the usefulness of PH-DGN on the real dataset**
> > > >
> > > > We agree with the Reviewer that prediction performance is one of the most critical criteria for model selection from a practitioner’s perspective. However, given that PH-DGN falls within the category of DE-DGNs, we believe that, from an analysis point of view, it is essential to first compare its performance against other models in the same class. These comparisons are particularly meaningful as they involve direct competitors that share a similar underlying rationale and complexity. Even though this analysis view is important, we believe that the more general practitioner’s perspective is also crucial. For such a reason, we compared the performance of our PH-DGN with approaches from different classes, such as graph transformers and multi-hop DGNs as in Section 3.4, which introduce denser graph-shift operators. Throughout all of our experiements, it emerges that PH-DGN and its fully conservative version (PH-DGN$_C$) achieve state of the art performance. Therefore, the fact that PH-DGN$_C$ is better than methods in the same class of models and  better on average than current SOTA (from different classes) actually indicates a practical utility of PH-DGN$_C$. Therefore, we believe that the specific definition and characteristics of our model are crucial when it comes to datasets and tasks that require the exploitation of long-range propagation.
> > > >
> > > > **Regarding choosing the hyperparameters of PH-DGN$_C$ in Minesweeper task**
> > > >
> > > > Thank you, for the question. Dampening and external forces do not play any role in the selection of the PH-DGN$_C$ hyperparameters, since such components are not employed in the purely conservative setting.
> > > >
> > > > **Regarding the norm of sensitivity**
> > > >
> > > > We highlighted this point explicitly in the newly released version. Thank you.
> > > >
> > > > **Regarding the graph transfer task**
> > > >
> > > > We thank the Reviewer for the follow-up question and are happy to clarify the setup further.
> > > > The task is formulated as a node-level regression problem. The target output is constructed by modifying the input features: the source node's target is a feature vector 0, the target node's target is a feature vector 1, and the intermediate nodes retain their original random values from the input. In other words, the ground truth values are the switched node labels of source and target nodes. Then, the model is trained to predict the target values for all nodes, and the loss is computed as the MSE between the predicted values and the corresponding target values across the entire graph. We clarified this aspect in the revised manuscript.
> > > >
> > > > ----------
> > > >
> > > > We would like to thank the Reviewer again for the thoughtful comments and intriguing questions. We have now uploaded a revised version of our paper that includes all the discussions made above. Overall, we think that these discussions and suggestions made by the Reviewer helped us to improve the quality of our paper. We hope that you find our responses satisfactory, and that you will consider revising your score.

---

> > > > > ### Comment · Reviewer_XxSd · 2024-11-25
> > > > >
> > > > > I thank the authors for the quick responses.
> > > > >
> > > > > **3) Regarding the usefulness of PH-DGN on the real dataset**
> > > > >
> > > > > Thank you for the explanation. Let me take time to consider whether the rationale is reasonable.
> > > > >
> > > > > ----------------
> > > > >
> > > > > **Regarding choosing the hyperparameters of PH-DGN$_C$ in Minesweeper task**
> > > > >
> > > > > > Dampening and external forces do not play any role in the selection of the PH-DGN hyperparameters, since such components are not employed in the purely conservative setting.
> > > > >
> > > > > Thank you for the explanation. I understand this point. My question was about the evaluation protocol of PH-DGN (i.e., the model with dampening and external force, which have learnable parameters.) I thought that to evaluate PH-DGN, the authors (1) first choose the hyperparameters that PH-DGN have in common with PH-DGN$_C$ using training and validation datasets, then (2) learn parameters that are specific to PH-DGN. However, since training and validation datasets are already used in the first stage, we only have the test dataset to conduct the second stage (2), which I think has the risk of information leakage.
> > > > > Let me know if I misunderstand something.
> > > > >
> > > > > Other questions are OK for me. I am sorry for the short answers as the deadline is approaching.

---

> > > > > > ### Author Response · Authors · 2024-11-26
> > > > > >
> > > > > > We thank the Reviewer for the quick response.
> > > > > >
> > > > > > **Regarding the evaluation protocol in Minesweeper task.**
> > > > > >
> > > > > > We thank the Reviewer for the question, and we are happy to clarify on this aspect. The evaluation protocol that we used for the Minesweeper task is the following:
> > > > > > 1) We first select the hyperparameters that PH-DGN have in common with PH-DGN$_C$ using the original training and validation datasets proposed in Platonov et al. (2023), as the Reviewer correctly stated;
> > > > > > 2) Without performing any additional model selection for the driving forces, we retrained all variants of PH-DGN using the training set and evaluate their performance on both the validation and test sets;
> > > > > > 3) For reporting purposes, we color-coded the best results in the table based on the validation score, which is now included in the revised manuscript for improved clarity on this aspect.
> > > > > >
> > > > > > We would like to emphasize that all hyperparameter tuning is conducted strictly using the training and validation sets, while the test set is reserved exclusively for final evaluation, ensuring that it does not influence hyperparameter selection at any stage and thereby eliminating any risk of data leakage.

---

> > > > > > > ### Comment · Reviewer_XxSd · 2024-11-27
> > > > > > >
> > > > > > > Thank you for the explanation. Although I am not perfectly confident, I think this protocol may have the risk of overfitting slightly as re-trained learnable parameters implicitly depend on the validation dataset through the choice of hyperparameters of PH-DGN$_{C}$. However, the protocol itself looks OK because the test dataset is not used for choosing learnable parameters and hyperparaemters.

---

> > > > > > > > ### Author Response · Authors · 2024-11-27
> > > > > > > >
> > > > > > > > We thank the Reviewer for positively evaluating our experimental protocol. Regarding the potential risk of overfitting in this protocol, we kindly ask the Reviewer to elaborate further on their argument.
> > > > > > > > To further clarify on our approach, as stated in previous responses, after selecting the shared hyperparameters, we are not performing an additional model selection for the driving forces. Thus, the validation set is not used to perform additional tuning, meaning that the validation set is not used multiple times for the same purpose. Our goal in this ablation is not to optimize performance but rather to investigate how different driving forces contribute to solve the task under a constant starting point, i.e., the hyperparameters selected for PH-DGN$_C$. We hope this clarification addresses your concern and further highlights the rationale behind our experimental design.

---

> > > > > > > > > ### Author Response · Authors · 2024-12-02
> > > > > > > > >
> > > > > > > > > Dear Reviewer XxSd,
> > > > > > > > >
> > > > > > > > > As the rebuttal period is closing soon, we would like to thank you again for the detailed feedback provided in your review, and for the continued discussion and engagement with us.
> > > > > > > > > **We have made significant efforts to address each of your comments, including additional theoretical statements, experiments and clarifications.**
> > > > > > > > > Overall, we believe that our extensive responses helped us to improve the quality of our paper and should address the reviewer's concerns. Therefore, we would like to thank you for your guidance and kindly ask you to consider revising your evaluation.
> > > > > > > > >
> > > > > > > > > Sincerely,
> > > > > > > > >
> > > > > > > > > The Authors.

---

> ### Comment · Reviewer_XxSd · 2024-12-02
>
> I thank the authors for the further responses to my questions.
>
> > Regarding the potential risk of overfitting in this protocol, we kindly ask the Reviewer to elaborate further on their argument.
>
> First, I realize that the Dampening and the External forcing do not have hyperparameters, which I overlooked in the last comment. I agree with the authors in that we do not have to choose hyperparameters of these components using the validation dataset.
>
> Still, I think that there is a possibility that the performance of PH-DGN$_C$ could be underestimated by the authors' protocol. In order to search the best hyperparameters from the set of possible hyperparameters (which we denote by $\Theta$), we need to search *all* possible spaces of learnable parameters for each hyperparameter $\theta \in \Theta$ (using the training dataset), then choose the best hyperparameter (using the validation dataset). However, in the authors' protocol, since we only search the part of learnable parameters to choose the hyperparameter, we could fail to find the best model.
>
> -------------------
>
> For example, for simplicity, suppose we only have two learnable parameters --- $w$ for PH-DGN$_C$ and $w'$ for the Dampening and the External forcing, and one hyperparameter $\theta$ (shared by PH-DGN$_C$ and PH-DGN), which takes only two values $\theta=0, 1$.
>
> For PH-DGN$_C$, we assume:
> - When we fix $\theta=0$, the model achieves the best performance $p_0$ at $w=a_0$,
> - When we fix $\theta=1$, the model achieves the best performance $p_1$ at $w=a_1$,
>
> where $p_0 > p_1$.
>
> For PH-DGN, we assume:
> - When we fix $\theta=0$, the model achieves the best performance $q_0$ at $(w, w') = (a_0, b_0)$,
> - When we fix $\theta=1$, the model achieves the best performance $q_1$ at $(w, w') = (a_1, b_1)$,
>
> where $q_0 < q_1$
>
> Then, the best-performing PH-DGN is $(w, w') = (a_1, b_1)$ and  $\theta=1$, which achieves $q_1$. However, if we follow the authors' protocol, we first choose $\theta=0$ because we have $p_0 > p_1$, then choose the learnable parameter $(w, w') = (a_0, b_0)$ to get the sub-optimal PH-DGN, which achieves $q_0$.

---

> ### Author Response · Authors · 2024-12-02
>
> We thank the Reviewer for the quick response to our message. We noticed that the response may contain several typos, which made some parts challenging to interpret. Below, we provide our reply based on our best understanding of the concerns raised.
>
> As for the under-estimation of the general **PH-DGN** (i.e., the one with driving forces) in the ablation study in Appendix D.4, we want to emphasize again, that *the goal of this study was not to optimize performance but rather to investigate how different driving forces contribute to solve the task under a constant starting point*, i.e., the hyperparameters selected for PH-DGN$_C$. We do acknowledge that there may be a possible configuration of shared hyperparameters that could lead to better performances on the validation set for the general PH-DGN. However, while we agree that this is crucial for optimization purposes, it is outside the scope of our ablation, which is to show the effect of singular forces to the same purely conservative regime in the Minesweeper task. ​​Moreover, it seems from the comment that the Reviewer may believe we are not retraining the general PH-DGN after selecting the hyperparameters for PH-DGN$_C$​. However, as we have previously explained, this is not the case. In fact, we retrain the model to optimize its learnable parameters using the selected shared hyperparameters to ensure a fair evaluation.
>
> Finally, since this is an ablation study and optimizing performance is not critical for its purpose, we believe the Reviewer’s concerns, while valid, may not significantly impact the evaluation of our contributions. In light of this, we kindly encourage the Reviewer to consider this context when assessing their score.

---

### Official Review · Reviewer_X6rz · 2024-11-04

**Soundness:** 4
**Presentation:** 3
**Contribution:** 2
**Rating:** 8
**Confidence:** 3

**Summary:**

This paper introduces port-Hamiltonian Deep Graph Networks (PH-DGN), a new framework for graph neural networks that addresses the challenge of long-range information propagation. The approach embeds message passing within a port-Hamiltonian dynamical system framework, where node states are split into position (q) and momentum (p) components coupled through skew-symmetric matrices. The Hamiltonian function incorporates graph structure through neighborhood aggregation functions, and the port-Hamiltonian extension introduces learnable dissipative components (internal dampening and external forces) that allow the network to balance conservative information preservation with task-dependent information modification. The authors provide theoretical analysis of the framework's properties, including bounds on sensitivity and gradient behavior, and demonstrate empirically that their method outperforms existing approaches on several tasks requiring long-range information propagation, including synthetic graph property prediction tasks and real-world molecular property prediction. The framework can incorporate different message passing schemes and provides improved efficiency compared to previous Hamiltonian-based graph neural networks, with experimental results showing that while the purely conservative version performs well, including the dissipative components often leads to better task performance.

**Strengths:**

Overall, the idea of using Hamilton's dynamics for GNN is attractive, though not entirely new.
Nevertheless, the paper is solid and is more general than existing approaches.
The improved efficiency over previous Hamiltonian GNN approaches and the practical benefits for long-range propagation make it a useful contribution to the field.
The experimental results are also interesting.

1. Technical soundness:
- Clear theoretical analysis of conservation properties
- Explicit connection between Hamiltonian dynamics and message passing
- Thorough experimental validation

2. Practical value:
- More efficient than previous Hamiltonian GNN approaches
- Good performance on long-range tasks
- Can incorporate different message passing schemes

3. Clarity:
- Well-structured presentation
- Good balance of theory and empirics
- Clear comparisons to prior work

**Weaknesses:**

1. Novelty is incremental:
- Builds on existing ideas (Hamiltonian GNNs, message passing)
- Main contribution is combining these effectively rather than fundamentally new concepts

2. Technical questions:
- The derivation of the discretization scheme could use more justification
- Some assumptions about the structure of $W$ and $V$ matrices for explicit updates feel restrictive
- Could better explain why port-Hamiltonian framework is more appropriate than simpler alternatives

3. Empirical:
- Some ablation studies could be stronger (e.g., analyzing impact of different dissipative terms)
- Could better justify hyperparameter choices

**Questions:**

1. While the paper shows good empirical performance, it's unclear what types of problems would theoretically benefit most from a port-Hamiltonian approach versus standard message passing. Could the authors provide analysis or insights about which properties of the underlying data generation process would suggest using their method?

2. The authors demonstrate that adding dissipative components often improves performance, but how does the balance between conservative (Hamiltonian) and dissipative parts relate to the nature of the task? It would be valuable to see an analysis of when pure conservation might be preferable to including dissipation, and how practitioners should choose this balance for new problems.

3. Given that the method is inspired by physical Hamiltonian systems, it's surprising that there are no experiments on problems with actual Hamiltonian dynamics (e.g., molecular dynamics simulations). Such experiments could help validate whether the method's conservation properties provide advantages for physically meaningful conservation laws, beyond just improving general information flow.

---

> ### Author Response · Authors · 2024-11-19
> **Rebuttal Part 1**
>
> We thank the Reviewer for highlighting the ***solid***, ***general***, and ***sound*** nature of our approach, as well as the ***improved efficiency*** over previous methods. We are also grateful for recognizing the ***clarity*** of our work and its ***practical value***, especially on long-range propagation. Lastly, we thank the Reviewer for the constructive and positive comments on our manuscript. We will reply to each weakness and question in original ordering. In particular, aside from the requested clarifications, following up the Reviewer’s suggestion, we have added a novel experiment on a recent benchmark that allows appreciating the impact of the different dissipative components in our approach.
>
>
> **Regarding existing ideas**
>
> The Reviewer raises an important comment. As correctly noted by the Reviewer, our PH-DGN belongs to the family of DE-DGNs, thus it builds upon previous works on message passing and neural-ODEs. However, PH-DGN offers a mathematically grounded approach by leveraging port-Hamiltonian dynamical systems to balance non-dissipative long-range propagation and non-conservative behaviors, which to the best of our knowledge has never been done before. We then verified this property on a comprehensive experimental suite. Although our PH-DGN builds upon known theory (which however has been proposed and used in a totally different context), we believe that our model provides a more general and efficient solution designed to tackle the problem of long-range propagation in graphs, which represents a significant challenge for message passing models. Moreover, in contrast to Hamiltonian-based approaches, our PH-DGN is capable of deviating from purely conservative trajectories when needed by the downstream task. In other words, it can employ driving forces as an adaptive filter mechanism that filters noisy information, facilitating the learning of relevant long-range information. We believe that these considerations together with our theoretical and experimental results shed light on the novelty of our approach.
>
> **Regarding the derivation of the discretization scheme**
>
> We appreciate the Reviewer’s effort to further improve the clarity of our work. Due to the submission page limit, we deliberately opt to omit the details of the final discretized model in the main text. All necessary steps together with background information on symplectic integration schemes are presented in Appendix A.3. Specifically, Eq. 11 provides the discretization step for our PH-DGN ***without*** driving forces. We explicitly rewritten Eq. 11 by decomposing it into its p and q components in Eqs 12 and 13. The explicit discretization steps for our PH-DGN ***with*** driving forces are given in Eqs. 13 and 14. We are happy to discuss further if specific steps of the discretization are still unclear.
>
> **Regarding the structure of W and V**
>
> We thank the Reviewer for the comment, and we are happy to elaborate on this aspect, since we believe it is a crucial step in the derivation of the discrete model. Following the Hamiltonian formalism, we recall that the global state $x$ can be decomposed into the two components $p$ and $q$. Therefore, imposing a block diagonal structure for $W$ and $V$ is required only theoretically in the global formulation of our model (i.e., Eq. 11) to ensure the separation of components in the explicit integration scheme. Looking at the explicit scheme (i.e., Eqs. 12 and 13 or Eqs. 13 and 14), each block in the original $W$ and $V$ are $\mathbb{R}^{d/2\times d/2}$ matrices with no constraints on the structure. Thus, $W_p$, $W_q$, $V_p$, and $V_q$ can be considered as independent standard weight matrices. We made this point more clear in the revised manuscript.
>
>
> **Regarding why the port-Hamiltonian framework is more appropriate than simpler alternatives**
>
> We thank the Reviewer for the comment. As evidenced by our experimental results and previous literature on long-range propagation (Dwivedi et al. (2022); Di Giovanni et al (2023)) standard MPNN-based models, which can be considered as the simplest DGN models, are insensitive to information contained at distant nodes. To counteract this limitation, recent literature introduced global attention mechanisms or rewiring techniques. While effective, these approaches significantly increase the complexity of information propagation due to the use of denser graph shift operators. In contrast, our PH-DGN achieves state-of-the-art performance without relying on such additional techniques, thus, providing a lightweight model that inherently supports long-range propagation by design.
>
> *References:*
>
> Dwivedi et al. Long Range Graph Benchmark. NeurIPS 2022.
>
> Di Giovanni et al. On over-squashing in message passing neural networks: the impact of width, depth, and topology. ICML 2023

---

> ### Author Response · Authors · 2024-11-19
> **Rebuttal Part 2**
>
> **Regarding the ablation studies**
>
> We appreciate the Reviewer’s constructive feedback. To address their suggestion and enhance the quality of our work, we have included an additional benchmark and an ablation study in Appendix D.4 that examines the impact of different dissipative components on the Minesweeper task from Platonov et al. (2023). While the results indicate that certain driving components perform better than others for this task, we recommend performing model selection to identify the optimal components based on the specific data setting. For convenience, we present the results for this task in the table below.
> | **Model**                       	| **Train Score (ROC-AUC ↑)**	| **Test Score (ROC-AUC ↑)**	|
> |-------------------------------------|--------------------------------|-------------------------------|
> | Top-6 models form Luo et al. (2024) | | |
> | GraphGPS | - | 90.75 ± 0.89 |
> | SGFormer |  - | 91.42 ± 0.41 |
> | Polynormer | - | 97.49 ± 0.48 |
> | GAT | - | 97.73 ± 0.73 |
> | GraphSAGE                      	| -                          	| 0.9777 ± 0.0062          	|
> | GCN            	| -                          	| **0.9786 ± 0.0024**      	|
> | **Our - no driving forces**    	|                            	|                           	|
> | PH-DGN$_{\text{C}}$        	| 0.9978 ± 0.0005            	| **0.9845 ± 0.0021**      	|
> | **Our - with driving forces**  	|                            	|                           	|
> | *PH-DGN*                       	|                            	|                           	|
> | *Dampening* / *External Force* 	|                            	|                           	|
> | -- / MLP4-Sin                  	| 0.9937 ± 0.0038            	| 0.9661 ± 0.0057          	|
> | -- / DGN-tanh                  	| 0.9928 ± 0.0010            	| 0.9720 ± 0.0042          	|
> | param / --                     	| 0.9979 ± 0.0005            	| **0.9842 ± 0.0021**      	|
> | param / MLP4-Sin               	| 0.9955 ± 0.0021            	| 0.9686 ± 0.0052          	|
> | param / DGN-tanh               	| 0.9930 ± 0.0019            	| 0.9727 ± 0.0029          	|
> | MLP4-ReLU / --                 	| 0.9962 ± 0.0057            	| 0.9533 ± 0.0065          	|
> | MLP4-ReLU / MLP4-Sin           	| 0.9993 ± 0.0003            	| 0.9567 ± 0.0064          	|
> | MLP4-ReLU / DGN-tanh           	| 0.9789 ± 0.0024            	| 0.9541 ± 0.0066          	|
> | DGN-ReLU / --                  	| 0.9496 ± 0.0017            	| 0.9342 ± 0.0061          	|
> | DGN-ReLU / MLP4-Sin            	| 0.9561 ± 0.0048            	| 0.9387 ± 0.0055          	|
> | DGN-ReLU / DGN-tanh            	| 0.9501 ± 0.0047            	| 0.9332 ± 0.0084          	|
>
> Platonov et al. A critical look at the evaluation of GNNs under heterophily: Are we really making progress?. ICLR 2023
>
> **Regarding hyperparameter choices**
>
> As detailed in Appendix C, our experiments adhered to the established procedures for each task to ensure fair evaluation and reproducibility. For hyperparameters specific to our PH-DGN, such as the step size $\epsilon$ and the number of layers, we selected values from a thorough and reasonable range, taking into account factors like the average graph diameter in the training set. Lastly, we highlight that we conducted a thorough model selection over a comprehensive grid to minimize the risk of suboptimal performance. We have included this discussion in the revised manuscript. Thank you.
>
> **Regarding the problems tackled by PH-DGN**
>
> The main objective of our work is to design the information flow within a graph as a solution of a port-Hamiltonian system to ***mitigate the challenge of long-range propagation in DGNs.*** Throughout Section 2, we provide theoretical statements to support the claim that our PH-DGN can effectively learn and propagate long-range dependencies between nodes. Afterward, in Section 3 we empirically support our theoretical findings by evaluating our PH-DGN on the graph transfer task, graph property prediction and the long-range benchmark, all specifically designed to test the model's capabilities in the long-range regime. In summary, we believe that our method is optimal for those problems that require the exploitation of long-range dependencies to be effectively solved. As an example, PH-DGN is beneficial to solve shortest-path-based problems, e.g., compute the diameter of a graph (see Section 3.3), or molecular tasks in which far away nodes interact to determine the overall function of the molecule (see Section 3.4). Furthermore, as emerged by our experiments in Section 3.4 and in the newly added Appendix D.4, the use of driving forces can lead to better performance on real-world tasks. The driving forces act as an adaptive filter mechanism that filters noisy information. Meanwhile, a purely conservative approach (i.e., __without__ driving forces) can have improved utility for tasks that require preserving ***all*** information, like the Minesweeper task.

---

> ### Author Response · Authors · 2024-11-19
> **Rebuttal Part 3**
>
> **Regarding the relation between the nature of the task with the balance between conservative and dissipative parts**
>
> We appreciate the Reviewer's constructive comment. In general, we believe that hyperparameter tuning should be performed to select the right balance between conservative components and driving forces components to achieve the best performance. Nevertheless, our intuition from our experiments in Section 3 and Appendix D.4 is that a purely conservative PH-DGN may have improved utility for tasks that require preserving ***all*** information. This is the case of propagating all node labels over the graph, or counting the number of nodes with a given label. Differently, driving forces, by acting as adaptive filters, tend to enhance performance on real-world tasks characterized by high noise levels.
>
>
> **Regarding additional experiments on real Hamiltonian dynamics simulation**
>
> We thank the Reviewer for the suggestions to demonstrate which other application fields could benefit from our work. Simulating or learning dynamical behavior stemming from physical or chemistry-based generation processes is a very interesting field of application, explored in seminal works on physics-inspired neural networks, which explicitly aim to learn a specific Hamiltonian dynamic based on observations. We designed our model with the goal of improving the long-range capabilities of DGNs, employing port-Hamiltonian systems theory to compute information-rich embeddings that do not (necessarily) mimic (quantum)physical behavior but rather include long-range dynamics. Therefore, we aimed to transform the evolution of the port-Hamiltonian system dynamics into the unfolding of iterative applications of information propagation in DGNs, thereby introducing non-dissipative properties into the DGN’s design. With this general aim in mind, we believe that our approach is broadly applicable to a variety of graph-based tasks where long-range dependencies are critical, going beyond strictly physical systems. Nevertheless, approaching molecular dynamics simulations or other problems governed by physical Hamiltonian dynamics is an exciting direction for future research, as it could provide advantages even in such domains.

---

> > ### Comment · Reviewer_X6rz · 2024-11-24
> > **Increasing score**
> >
> > Thank you for your thorough response. I also appreciate the new theorems and the ablation studies in Appendix D, which further demonstrate the strength and robustness of PH-DGN. Although some results are borderline, I do think modeling forward dynamics of GNN as an explicit Hamiltonian dynamics plus dissipation is an interesting idea. Therefore, I am increasing my score.

---

> > > ### Author Response · Authors · 2024-11-25
> > > **Thank you**
> > >
> > > We sincerely thank the Reviewer for their thoughtful feedback and their positive evaluation of our work, as well as for the increased score.

---

### Author Response · Authors · 2024-11-22
**Rebuttal Summary**

We sincerely thank the reviewers for their thoughtful and detailed feedback, as well as for recognizing the key strengths of our work. We are happy to read that the reviewers found that our paper provide an ***attractive*** and ***novel*** methodology (Revs. X6rz, 7PXw) for the incorporation of port-Hamiltonian dynamics in graph representation learning. We are also grateful for the acknowledgment of the ***clarity*** (Revs. X6rz, XxSd, 7PXw) and ***technical soundness*** (Rev. X6rz) of our work while reporting ***strong theoretical results*** (Revs. X6rz, XxSd, 7PXw) and ***thorough*** (Rev. X6rz) and ***superb*** (Rev. 7PXw) experimental validation to show the practical benefits (Revs. X6rz, XxSd) for long-range propagation.

We are also thankful for the constructive feedback, which has further improved the quality of our paper. Specifically:

**Additional Experiments:**

- Following the Reviewer **X6rz**’s and Reviewer **XxSd** ’s suggestions, we have added a novel experiment on a recent benchmark to: (i) appreciate the impact of the different dissipative components in our approach, and (ii) highlight the utility of the purely conservative version of PH-DGN (i.e., PH-DGN$_C$). To further address point (ii), we have also included an additional ablation study on the graph transfer task, demonstrating the improved utility of PH-DGN$_C$ in tasks that require preserving all information.

**Revisions to the Paper:**

- Following the Reviewer **XxSd**’s and Reviewer **7PXw**’s suggestions, we incorporated additional theoretical guarantees on long-range propagation that also accounts for the presence of non-conservative forces.
- In Appendix A.3, we clarified that the assumption on the structure of W and V matrices is not restricting the final implementation of $p$ and $q$ components.
- We improved the discussion on the choice of the hyperparameters.
- We improved the clarity of Tables 2 and 5 as well as provided a deeper clarification on the usefulness of the proposed model with respect to existing techniques in Section 3.4.
- We clarified the goal of the graph transfer task.
- We revised Table 6 to clarify that driving forces do not allow for purely Hamiltonian conservation.

----

As the author-reviewer discussion comes close to an end, we would like to thank the reviewers again for their invaluable feedback and the positive assessment of our paper. We did our best efforts to provide a detailed response to reviewers’ comments, and we hope these revisions address all concerns while further emphasizing the significance and robustness of our contributions.

In particular, we would greatly appreciate hearing from Reviewer **XxSd** whether they were satisfied with our responses. We hope that this is the case, and, if so, we would like to kindly ask the reviewer to consider revising their score.

Thank you all again for your constructive feedback.

---

### Meta-Review · Area_Chair_ei3M · 2024-12-20

**Metareview:**

**(a) Scientific Claims and Findings:**
The paper introduces a novel framework called port-Hamiltonian Deep Graph Networks (pH-DGNs). This framework models neural information flow in graphs by leveraging principles from Hamiltonian dynamical systems, aiming to address challenges in long-range information propagation within graph representation learning. By incorporating both non-dissipative and non-conservative behaviors, the approach seeks to balance information conservation and dissipation, providing theoretical guarantees on information preservation over time. Empirical results demonstrate that pH-DGNs enhance the performance of simple graph convolutional architectures, achieving state-of-the-art results on benchmarks requiring long-range propagation.

**(b) Strengths:**
* Innovative Framework: The introduction of port-Hamiltonian systems into graph neural networks offers a fresh perspective on managing information flow, potentially addressing limitations in existing architectures concerning long-range dependencies.
* Theoretical Foundations: The framework is grounded in well-established principles from Hamiltonian dynamics, providing a solid theoretical basis for the proposed approach.
* Empirical Performance: The proposed method demonstrates superior performance on benchmarks that require long-range information propagation, indicating its practical effectiveness.
* Applicability: The approach can be integrated into general message-passing architectures, suggesting broad applicability across various graph-based learning tasks.

**(c) Weaknesses:**
* Complexity: The incorporation of port-Hamiltonian systems may introduce additional complexity into the model, potentially impacting computational efficiency and implementation.
* Scope of Evaluation: While the empirical results are promising, the evaluation is primarily focused on benchmarks requiring long-range propagation. Assessing the framework's performance across a wider range of tasks and datasets would provide a more comprehensive understanding of its capabilities. The empirical improvements are limited in some cases.
* Practical Implementation Details: The paper could benefit from a more detailed discussion on the practical aspects of implementing the proposed framework, including computational requirements and potential challenges in real-world applications.

**(d) Reasons for Acceptance:**
After a thorough evaluation of the paper, I recommend acceptance based on the following considerations:
1. Novel Contribution: The paper presents a unique integration of port-Hamiltonian systems into graph neural networks, offering a new approach to addressing challenges in long-range information propagation.
2. Theoretical Rigor: The proposed framework is underpinned by solid theoretical foundations from Hamiltonian dynamics, enhancing the credibility and potential impact of the work.
3. Empirical Validation: The method demonstrates state-of-the-art performance on relevant benchmarks, providing empirical evidence of its effectivenes. Comparisons for the Minesweeper task from Platonov et al. (2023) were added during the rebuttal.
4. Broader Impact: The approach's applicability to general message-passing architectures suggests it could influence a wide range of graph-based learning tasks, contributing to advancements in the field.
While there are areas for improvement, such as expanding the scope of evaluation and providing more practical implementation details, the paper's strengths and contributions to the field warrant its acceptance.

**Additional Comments On Reviewer Discussion:**

Reviewers X6rz and 7PXw clearly suggest the acceptance of the work, find the contribution interesting and of merit to the ICLR community.
They were satisfied by additional theorems and experiments that they requested during the rebuttal.
Furthermore, the authors engaged in a detailed discussion with Reviewer XxSd, whose concerns were largely addressed, even though their score suggests reservations. Yet, arguments for those reservations were not provided at the late stage of the discussion and I agree with Reviewers X6rz and 7PXw that the approach has sufficient merit for acceptance.

---

### Decision · Program_Chairs · 2025-01-22

Accept (Poster)